# Fast Catch-Up, Late Switching: Optimal Batch Size Scheduling via Functional Scaling Laws

**Jinbo Wang**[1,*], **Binghui Li**[2,*], **Zhanpeng Zhou**[3], **Mingze Wang**[1], **Yuxuan Sun**[4],
**Jiaqi Zhang**[5,†], **Xunliang Cai**[5] **& Lei Wu**[1,2,6,†]

[1]School of Mathematical Sciences, Peking University
[2]Center for Machine Learning Research, Peking University
[3]School of Computer Science, Shanghai Jiao Tong University
[4]State Key Laboratory of Cognitive Intelligence, University of Science and Technology of China
[5]Meituan, Beijing  [6]AI for Science Institute, Beijing
[*]wangjinbo@stu.pku.edu.cn,libinghui@pku.edu.cn
[†]zhangjiaqi39@meituan.com,leiwu@math.pku.edu.cn

## Abstract

Batch size scheduling (BSS) plays a critical role in large-scale deep learning training, influencing both optimization dynamics and computational efficiency. Yet, its theoretical foundations remain poorly understood. In this work, we show that the **functional scaling law (FSL)** framework introduced in Li et al. (2025a) provides a principled lens for analyzing BSS. Specifically, we characterize the optimal BSS under a fixed data budget and show that its structure depends sharply on task difficulty. For easy tasks, optimal schedules keep increasing batch size throughout. In contrast, for hard tasks, the optimal schedule maintains small batch sizes for most of training and switches to large batches only in a late stage. To explain the emergence of late switching, we uncover a dynamical mechanism—the **fast catch-up** effect—which also manifests in large language model (LLM) pretraining. After switching from small to large batches, the loss rapidly aligns with the constant large-batch trajectory. Using FSL, we show that this effect stems from rapid forgetting of accumulated gradient noise, with the catch-up speed determined by task difficulty. Crucially, this effect implies that *large batches can be safely deferred to late training* without sacrificing performance, while substantially reducing data consumption. Finally, extensive LLM pretraining experiments—covering both Dense and MoE architectures with up to **1.1B** parameters and **1T** tokens—validate our theoretical predictions. Across all settings, late-switch schedules consistently outperform constant-batch and early-switch baselines.

## 1 Introduction

Large language model (LLM) pretraining demands massive computational resources, making training efficiency a central challenge. At scale, training efficiency depends critically on parallelism, and increasing the batch size directly improves hardware utilization and throughput (Goyal et al., 2017; Brown et al., 2020; Hoffmann et al., 2022). Large-batch training has therefore become indispensable for scalable LLM pretraining.

However, using a constant large batch size throughout training is suboptimal in terms of sample efficiency (McCandlish et al., 2018; Merrill et al., 2025). From a stochastic optimization perspective, the batch size determines the noise scale of stochastic gradients: each update can be viewed as the population gradient perturbed by noise whose variance decreases with the batch size. In the early stages of training, the optimization dynamics are signal-dominated, so aggressively reducing noise via large batches yields limited benefit while consuming more data. As training proceeds, the signal weakens and the influence of gradient noise increases, making larger batches more effective

---

[*]Equal contribution.
[†]Corresponding authors.

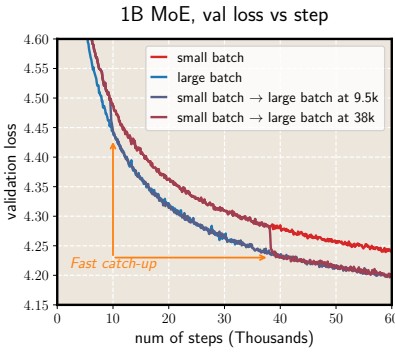 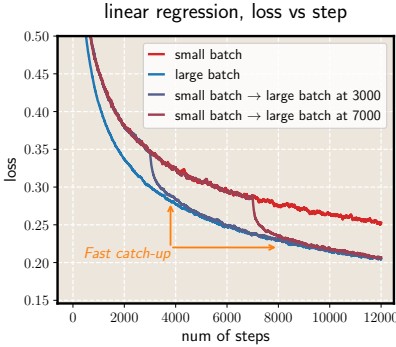

Figure 1: **The fast catch-up effect when switching from a small to a large batch size. Left:** Validation loss versus training steps for a 1B-parameter MoE model trained on approximately 0.4T tokens under four batch-size schedules: constant small batch, constant large batch, small-to-large with early switch, and small-to-large with late switch. **Right:** Validation loss versus training steps in the theoretical setting with $s = 0.3$ and $\beta = 1.5$ (the hard-task regime), which demonstrates the same catch-up effect.

for improving iteration efficiency. This motivates **batch size scheduling** (BSS), i.e., dynamically increasing the batch size during training.

Indeed, BSS has become ubiquitous in industrial-scale LLM pretraining, adopted in models such as GPT-3 (Brown et al., 2020), PaLM (Chowdhery et al., 2023), LLaMA-3 (Grattafiori et al., 2024), DeepSeek-V3 (DeepSeek-AI et al., 2024b), MiniMax-01 (MiniMax et al., 2025), Nemotron-4 (Parmar et al., 2024; Nvidia et al., 2024), and GLM-4.5 (Zeng et al., 2025). This widespread adoption calls for a principled understanding of how batch size scheduling shapes training dynamics and efficiency. Yet existing analyses either focus on constant batch sizes (Ma et al., 2018; Zhang et al., 2025) or rely on empirical and heuristic insights (Smith et al., 2018; McCandlish et al., 2018; Merrill et al., 2025). As a result, current BSS design often depends on heuristic tuning or expensive large-scale experimentation.

The functional scaling law (FSL) framework introduced in Li et al. (2025a) provides a continuous-time modeling of how batch size and learning rate schedules affect loss dynamics. While originally derived for linear regression and kernel regression, FSL exhibits strong expressive power for modeling the loss dynamics of practical LLM pretraining. However, Li et al. (2025a) focuses solely on learning rate schedules. In this paper, we extend this framework to analyzing BSS. Our contributions are as follows.

- **Optimal batch size schedule.** Under the FSL framework, we derive the optimal batch size schedule under a fixed data (or compute) budget. The optimal BSS depends critically on task difficulty: easy tasks favor a monotonically increasing schedule, while hard tasks require keeping batch sizes small for most of training, with growth deferred to a late phase. This stable-growth strategy increases the number of optimization steps under a fixed data budget, which benefits hard tasks. Extending to practical few-stage schedules, we find that easy tasks again favor constant large-batch training, whereas hard tasks demand a prolonged small-batch phase followed by a late switch to large batches.

- **The fast catch-up effect.** To explain why large batches can be safely deferred for hard tasks, we uncover a striking and highly robust *fast catch-up effect*: when training switches from a small to a large batch size, the loss rapidly collapses to that of the constant large-batch run. This phenomenon appears consistently across LLM pretraining experiments with diverse architectures, model scales, and data regimes (see Figure 1). Using FSL, we further provide a theoretical explanation of this effect and quantitatively characterize how task difficulty governs the speed of catch-up.

- **Large-scale validation of late-switch superiority.** The fast catch-up effect implies that large batches can be safely deferred to late training without sacrificing performance, while substantially reducing data consumption. We validate this principle through extensive LLM pretraining

experiments spanning Dense and MoE architectures, model sizes from 50M to **1.1B** parameters, and data scales up to **1T** tokens. Across all settings, stage-wise BSS with late switching consistently outperforms constant-batch and early-switch baselines.

## 1.1 RELATED WORK

**Neural scaling laws.** Hestness et al. (2017) first observed that the performance of deep learning follows predictable power-law relationships with model and data size, a phenomenon later formalized as *neural scaling laws* (Kaplan et al., 2020). These laws have since become guiding principles for configuring large-scale training and been refined across architectures and training regimes (Henighan et al., 2020; Hoffmann et al., 2022; Kadra et al., 2023; Aghajanyan et al., 2023; Muennighoff et al., 2023; Tissue et al., 2024; Luo et al., 2025; Qiu et al., 2025), with parallel theoretical efforts explaining their origins and mechanisms (Bordelon et al., 2024; Lin et al., 2024; Bahri et al., 2024; Paquette et al., 2024; Yan et al., 2025; Kunstner & Bach, 2025; Li et al., 2026a). In this work, we build on the framework of Li et al. (2025a) to provide a scaling-law analysis of batch size scheduling.

**Large-batch training and batch size scheduling.** Large-batch training is essential for leveraging hardware parallelism at scale. Existing work largely focuses on *static* batch sizes, aiming to determine how large the batch size can be increased without sacrificing data efficiency, typically characterized by the critical batch size (McCandlish et al., 2018; Ma et al., 2018; Kaplan et al., 2020; Gray et al., 2024; Zhang et al., 2025; Merrill et al., 2025). In practice, however, LLM pretraining routinely employs *batch size schedules*. Despite its prevalence, BSS has received far less theoretical attention than learning rate schedules (Defazio et al., 2023; Hu et al., 2024; Hägele et al., 2024). Existing analyses of BSS either rely on heuristic arguments (Smith et al., 2018; McCandlish et al., 2018) or are framed as optimal control problems (Lee et al., 2022; Zhao et al., 2022; Perko, 2023), offering limited structural insight. In contrast, we develop a scaling-law-based theory of BSS that systematically explains empirical practice and yields new design principles.

**One-pass SGD in kernel regression.** The convergence of one-pass stochastic gradient descent (SGD) in kernel regression—often interpreted as high-dimensional linear regression—has been extensively studied. In particular, Dieuleveut & Bach (2015); Mücke et al. (2019) showed that *averaged* one-pass SGD achieves the minimax-optimal rate $D^{-s\beta/(s\beta+1)}$ in easy-task regimes and the rate $D^{-s}$ in hard-task regimes. Subsequent work further established that the same rates can be attained by *last iterate* when combined with appropriate learning rate decay (Wu et al., 2022a; Lin et al., 2024; Li et al., 2026b). In contrast, we show that one-pass SGD with a *constant* learning rate, when coupled with a properly designed BSS, achieves the same optimal rates.

## 2 PRELIMINARIES

**Notation.** Throughout the paper, the notation $\asymp$ indicates equivalence up to a constant factor, and $\lesssim$ (resp. $\gtrsim$) indicates inequality up to a constant factor. For two nonnegative functions $f, g : \mathbb{R}_{\geq 0} \to \mathbb{R}_{\geq 0}$, we write $f(t) \asymp g(t)$ if there exist constants $C_1, C_2 > 0$, independent of $t$, such that $C_1 f(t) \leq g(t) \leq C_2 f(t)$, $\forall t \geq 0$.

## 2.1 FEATURE-SPACE LINEAR REGRESSION

Let $\mathcal{X}$ and $\mathcal{D}$ denote the input domain and distribution, respectively. Labels are generated as $y = f^\star(\boldsymbol{x}) + \epsilon$ with $\epsilon \sim \mathcal{N}(0, \sigma^2)$. We assume $\sigma \gtrsim 1$ and the target function $f^\star$ is given by $f^\star(\boldsymbol{x}) := \langle \phi(\boldsymbol{x}), \boldsymbol{\theta}^\star \rangle$. Here, $\phi : \mathcal{X} \to \mathbb{R}^N$ is a feature map and $\boldsymbol{\theta}^\star \in \mathbb{R}^N$ (with $N \in \mathbb{N}_+ \cup \{\infty\}$) is the unknown target parameter. We assume $\phi(\boldsymbol{x}) \sim \mathcal{N}(\boldsymbol{0}, \mathbf{H})$ with $\{\lambda_j\}_{j=1}^N$ denoting the eigenvalues of $\mathbf{H} := \mathbb{E}_{\boldsymbol{x} \sim \mathcal{D}} [\phi(\boldsymbol{x})\phi(\boldsymbol{x})^\top]$ in a decreasing order.

**Assumption 2.1** (Power-law structures). The following two conditions hold:

- **(Capacity condition)** $\lambda_j \asymp j^{-\beta}$ for some $\beta \in (1, \infty)$.

- **(Source condition)** $|\theta_j^\star|^2 \asymp j^{-1}\lambda_j^{s-1} = j^{-[1+(s-1)\beta]}$ for some $s \in (0, \infty)$.

The *capacity exponent* $\beta$ controls the decay rate of the eigenvalues. Smaller $\beta$ corresponds to a larger effective rank of the spectrum and thus higher model capacity. The *source exponent* $s$ measures the alignment of the target function with the kernel eigenstructure: smaller $s$ corresponds to harder learning problems, with more energy concentrated in high-frequency components. These capacity and source conditions are standard in the analysis of kernel methods and have recently been adopted in scaling-law studies (Paquette et al., 2024; Lin et al., 2024; Bordelon et al., 2025; Li et al., 2025a). A more detailed interpretation of the above setup is provided in Appendix A.1.

**One-pass SGD.** We learn the target function $f^\star$ using a student model $f(\boldsymbol{x}; \boldsymbol{\theta}) := \langle \phi(\boldsymbol{x}), \boldsymbol{\theta} \rangle$ by minimizing the population risk $\mathcal{R}(\boldsymbol{\theta}) := \frac{1}{2}\mathbb{E}_{\boldsymbol{x} \sim \mathcal{D}}[(f(\boldsymbol{x}; \boldsymbol{\theta}) - y)^2]$ via one-pass SGD. At each iteration $1 \le k \le K$, SGD samples a mini-batch $\mathcal{S}_k = \{(\boldsymbol{x}_{k,i}, y_{k,i})\}_{i=1}^{B_k}$ and performs the update

$$\boldsymbol{\theta}_{k+1} = \boldsymbol{\theta}_k - \frac{\eta}{B_k} \sum_{i=1}^{B_k} \nabla_{\boldsymbol{\theta}} \left[ \frac{1}{2} \left( f(\boldsymbol{x}_{k,i}; \boldsymbol{\theta}_k) - y_{k,i} \right)^2 \right], \tag{1}$$

where $\eta > 0$ is a constant learning rate, and $(B_1, B_2, \cdots, B_K)$ denotes the **batch size schedule (BSS)** satisfying the minimum batch size constraint $B_i \ge B_{\min} > 0$. Notably, the update in (1) can be rewritten as

$$\boldsymbol{\theta}_{k+1} = \boldsymbol{\theta}_k - \eta \left( \nabla \mathcal{R}(\boldsymbol{\theta}_k) + \boldsymbol{\xi}_k \right), \tag{2}$$

where $\boldsymbol{\xi}_k$ denotes the gradient noise that follows $\mathbb{E}[\boldsymbol{\xi}_k] = 0, \mathbb{E}[\boldsymbol{\xi}_k \boldsymbol{\xi}_k^\top] = \Sigma(\boldsymbol{\theta}_k)/B_k$, where $\Sigma(\boldsymbol{\theta})$ represents the covariance of gradient noise at $\boldsymbol{\theta}$ with the batch size 1. The learning performance is measured using the excess risk: $\mathcal{E}(\boldsymbol{\theta}) := \mathcal{R}(\boldsymbol{\theta}) - \frac{1}{2}\sigma^2 = \frac{1}{2}\|\boldsymbol{\theta} - \boldsymbol{\theta}^\star\|_{\mathbf{H}}^2$, where $\|\boldsymbol{v}\|_{\mathbf{H}}^2 := \boldsymbol{v}^\top \mathbf{H} \boldsymbol{v}$.

## 2.2 FUNCTIONAL SCALING LAWS

We analyze the loss dynamics of SGD using a continuous-time stochastic differential equation (SDE) model. The discrete update (2) can be modeled by the following Itô SDE (Li et al., 2019; Orvieto & Lucchi, 2019; Ankirchner & Perko, 2024):

$$\mathrm{d}\bar{\boldsymbol{\theta}}_t = -\nabla \mathcal{R}(\bar{\boldsymbol{\theta}}_t) \, \mathrm{d}t + \sqrt{\frac{\eta}{b(t)}\Sigma(\bar{\boldsymbol{\theta}}_t)} \, \mathrm{d}\boldsymbol{W}_t, \tag{3}$$

where $\boldsymbol{W}_t \in \mathbb{R}^N$ is an $N$-dimensional Brownian motion, and $b \in C(\mathbb{R}_{\ge 0})$ is the continuous-time batch size schedule with $b(k\eta) = B_k$ for all $k \in \mathbb{N}$. Here $t = kh$ represents continuous training time with discretization step size $h$, with each discrete iteration $k$ corresponding to time $t = k\eta$.

For the SDE (3), Li et al. (2025a) derived a functional scaling law (FSL) that characterizes the loss dynamics in continuous training time:

**Theorem 2.2** (Functional Scaling Law). *Under Assumptions 2.1, for sufficiently large $t$,*

$$\mathbb{E}[\mathcal{E}(\bar{\boldsymbol{\theta}}_t)] \asymp \underbrace{t^{-s}}_{\text{signal learning}} + \underbrace{\eta\sigma^2 \int_0^t \frac{\mathcal{K}(t - \tau)}{b(\tau)} \, \mathrm{d}\tau}_{\text{noise accumulation}}, \tag{4}$$

*where $\mathcal{K}(t) := (t + 1)^{-(2 - 1/\beta)}$.*

The above theorem is a simplification of Li et al. (2025a, Theorem 4.1) for constant learning rate. For completeness, we provide a self-contained derivation of the above FSL in Appendix A.2. **This law establishes a functional-level map from the BSS function to the loss at time $t$.** Notably, the two terms exhibit a clean interpretation:

- The signal-learning term corresponds to the learning under full-batch gradient descent, capturing the rate at which SGD extracts the signal $f^\star$. This rate is determined by the source exponent $s$.

- The noise-accumulation term characterizes how the BSS shapes the dissipation of gradient noise. The forgetting kernel $\mathcal{K}(t - \tau)$ characterizes how the noise injected at time $\tau$ still affects the loss at time $t$. Due to $\mathcal{K}(t) = (t + 1)^{-(2 - 1/\beta)}$, a higher-capacity model (smaller $\beta$) tends to forget noise more slowly.

While the FSL framework was introduced by Li et al. (2025a), their analysis was restricted to constant batch sizes and focused primarily on the analysis of learning rate scheduling. We extend this framework by showing that FSL also provides a principled tool for analyzing how batch size scheduling influences optimization dynamics and training efficiency.

## 3 THEORETICAL ANALYSES VIA FUNCTIONAL SCALING LAWS

We begin by asking the following question:

> *Given a total data budget $D$, what is the optimal batch-size schedule (BSS) when the loss dynamics follows the FSL (4)?*

For a fixed model, the data budget is equivalent to a *compute budget*, since the computational cost scales linearly with data size. Determining the optimal BSS is challenging, as the final-step loss depends on the entire training trajectory. This is essentially an optimal control problem (Zhao et al., 2022; Perko, 2023), which generally does not admit explicit solutions. However, the explicit characterization provided by FSL enables an analytical treatment of this problem. We address the above question under two settings: (1) unconstrained schedules; and (2) stage-wise BSS motivated by practical constraints.

### 3.1 OPTIMAL BATCH SIZE SCHEDULING WITHOUT SHAPE CONSTRAINTS

Under the FSL framework, seeking the optimal BSS can be formulated as solving the following resource-constrained *variational problem*:

$$
\min_{T>0,\, b(\cdot)} \quad \mathcal{E}_D[T, b] := \frac{1}{T^s} + \int_0^T \frac{\mathcal{K}(T-t)}{b(t)} \,\mathrm{d}t
$$

$$
\text{s.t.} \quad \int_0^T b(t)\,\mathrm{d}t = D, \qquad\qquad \text{(data/compute constraint)}
$$

$$
b(t) \geq B_{\min}, \quad \forall\, t \in [0, T], \qquad \text{(hardware constraint)}.
$$

(5)

Here, the integral constraint $\int_0^T b(t)\,\mathrm{d}t = D$ comes from the available data budget. The pointwise constraint $b(t) \geq B_{\min}$ captures hardware limitations in data-parallel training: the global batch size must be no smaller than the number of parallel devices (Narayanan et al., 2021).

We denote by $b^\star(\cdot)$ the optimal BSS for problem (5), and let $T^\star$ be the corresponding total training time. We further define the final-step loss as $\mathcal{E}_D^\star := \mathcal{E}_D[T^\star, b^\star]$.

**Theorem 3.1** (Optimal batch size schedule). *Assume $D$ and $B_{\min}$ are sufficiently large. Then:*

- **Easy-task regime** ($s > 1 - 1/\beta$). *The optimal BSS satisfies*

$$
b^\star(t) = B_{\max}\big(T^\star - t + 1\big)^{\frac{1}{2\beta}-1}, \qquad 0 \leq t \leq T^\star,
$$

*with $B_{\max} \asymp D^{\frac{1/2+s\beta}{1+s\beta}}$, $T^\star \asymp D^{\frac{\beta}{1+s\beta}}$. Moreover,*

$$
\mathcal{E}_D^\star \asymp D^{-\frac{s\beta}{1+s\beta}}.
$$

- **Hard-task regime** ($s \leq 1 - 1/\beta$). *The optimal BSS exhibits a two-phase stable-growth structure:*

$$
b^\star(t) = \begin{cases} B_{\min}, & 0 \leq t < T_1^\star, \\ B_{\max}\big(T^\star - t + 1\big)^{\frac{1}{2\beta}-1}, & T_1^\star \leq t \leq T^\star, \end{cases}
$$

*where $T^\star \asymp D$, $\frac{T^\star - T_1^\star}{T^\star} \asymp D^{-\frac{1-1/\beta-s}{2-1/\beta}}$, $B_{\max} \asymp D^{\frac{s+1}{2}}$. Moreover,*

$$
\mathcal{E}_D^\star \asymp D^{-s}.
$$

**The shape of the optimal BSS.** In the easy-task regime, the optimal BSS takes the form $b^\star(t) \approx B_{\max}(T^\star - t + 1)^{-\gamma}$, corresponding to a progressively increasing batch size throughout training, as illustrated in Figure 2 (left). The peak batch size scales with the data budget as $B_{\max} \approx D^\alpha$ with $\alpha > 0$, indicating that *larger datasets favor larger batch sizes*. This provides a theoretical explanation for the empirical practice of increasing batch size with dataset size (DeepSeek-AI et al., 2024a; Zhang et al., 2025; Li et al., 2025b).

In the hard-task regime, the optimal BSS exhibits a stable–growth structure: it stays at the minimal batch size $B_{\min}$ for the first $T_1^\star$ steps, followed by a growth phase with the same functional form as in the easy-task regime. Notably, for strictly hard-task regime $s < 1 - 1/\beta$, the growth phase occupies only a tiny fraction of the total training horizon, $(T^\star - T_1^\star)/T^\star = o_D(1)$, implying that *extremely large batch sizes are required only near the end of training*. See Figure 2 (left) for an illustration. Intuitively, for hard tasks (small $s$), maintaining a small batch size allows more optimization steps (larger $T$) under a fixed data budget, thereby significantly reducing the signal-learning term $T^{-s}$. The late-stage batch growth primarily serves to noise reduction. This stable–growth structure can be viewed as the batch-size analogue of the warmup–stable–decay learning rate schedule (Hu et al., 2024; Hägele et al., 2024; Li et al., 2026b).

**Same data efficiency, fewer iterations.** In the easy-task regime, the excess risk rate $D^{-s\beta/(1+s\beta)}$ matches the minimax optimal rate of this problem (Caponnetto & De Vito, 2007, Theorem 2). In the hard-task regime, the excess risk scales as $D^{-s}$, matching the best rate attainable by one-pass SGD (Dieuleveut & Bach, 2016; Pillaud-Vivien et al., 2018). These suggest that, with a properly designed BSS, constant learning rate can achieve the same data efficiency as carefully tuned learning rate schedules (Lin et al., 2024; Li et al., 2025a). The key distinction, however, lies in *iteration complexity*: batch-size scheduling significantly reduces the total number of iterations compared to learning rate scheduling. When coupled with modern GPU parallelism, this reduction directly translates into shorter wall-clock training time. In short, *batch-size scheduling preserves data efficiency while substantially reducing iteration complexity*.

**Numerical validation.** Although the FSL (4) is derived from the continuous-time SDE (3), we empirically confirm that Theorem 3.1 accurately predicts the behavior of discrete-time SGD. Specifically, we run SGD (1) using the optimal BSS prescribed by Theorem 3.1 and report the final-step loss as a function of the data size in Figure 2 (middle, right); see Appendix B.2 for experimental details. The observed data scaling closely matches the theoretical predictions. Additionally, in Appendix B.6.1, we compare constant learning rates combined with the optimal BSS against popular learning rate schedules (cosine and warmup–stable–decay). We find that the optimal BSS with a constant learning rate achieves comparable performance to these widely used learning rate schedules.

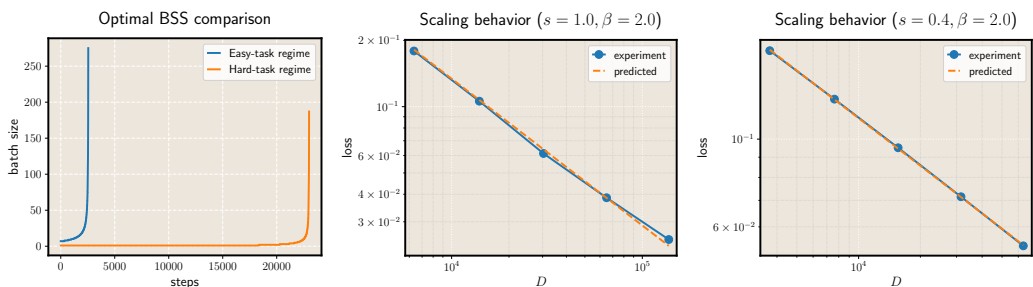

Figure 2: Optimal BSS experiments for the feature-space linear regression. **Left:** Illustration of the optimal BSSs for the easy-task and hard-task regimes. **Middle:** In the easy-task regime ($s = 1.0, \beta = 2.0$), one-pass SGD with optimal BSS attains the predicted minimax rate $D^{-s\beta/(1+s\beta)}$. **Right:** In the hard-task regime ($s = 0.4, \beta = 2.0$), it matches the optimal rate $D^{-s}$ attainable by one-pass SGD.

## 3.2 STAGE-WISE OPTIMAL BATCH SIZE SCHEDULING

Theorem 3.1 shows that the unconstrained optimal BSS follows a smoothly increasing schedule. In practice, however, batch sizes are discrete and constrained by hardware limitations. Moreover,

changing the batch size during training incurs nontrivial system overhead, such as data pipeline and communication reconfiguration. Consequently, practical schedules typically permit only a small number of stage-wise adjustments (DeepSeek-AI et al., 2024b; MiniMax et al., 2025).

In this section, we study the simplest nontrivial stage-wise setting: a two-stage schedule that begins with a small batch size $B_1$ and later switches to a larger batch size $B_2$. In practice, $B_1$ and $B_2$ are largely determined by hardware constraints, the key issue is to determine the optimal timing of this switch.

We denote by $\mathcal{E}_{B_1 \to B_2}(t)$ the loss at time $t$ under this two-stage schedule. Let $D$ be the total number of training samples and $P \in [0, D]$ denote the number of samples processed before switching from $B_1$ to $B_2$. The corresponding BSS $b_{B_1 \to B_2}^P(t)$, the switching time $T_{s,P}$, and the total training time $T_P$ are defined as follows:

$$b_{B_1 \to B_2}^P(t) = \begin{cases} B_1, & 0 \le t \le T_{s,P}, \\ B_2, & T_{s,P} < t < T_P, \end{cases} \qquad T_{s,P} = \frac{P}{B_1}, \quad T_P = \frac{P}{B_1} + \frac{D - P}{B_2}. \tag{6}$$

We denote by $\mathcal{E}_{B_1 \to B_2}^D(P)$ the expected final-step loss under this schedule.

**Theorem 3.2** (Optimal two-stage batch size schedule). *Let $B_1 < B_2$ be constants independent of $D$, and assume $D$ is sufficiently large. Define $P_D^\star = \arg\min_{P \in [0,D]} \mathcal{E}_{B_1 \to B_2}^D(P)$. Then:*

- *If $s > 1 - 1/\beta$, then $P_D^\star = 0$.*
- *If $s \le 1 - 1/\beta$, then $\frac{D - P_D^\star}{D} \asymp D^{-\frac{1 - 1/\beta - s}{2 - 1/\beta}}$.*

This theorem shows that, even within the restricted class of two-stage BSS, the optimal strategy still depends sharply on task difficulty. For easy tasks, it is optimal to employ large-batch training throughout. In contrast, for hard tasks with $s < 1 - 1/\beta$, one should maintain a small batch size for most of training and switch to a large batch only at a very late stage as $(D - P_D^\star)/D = o_D(1)$, consistent with the behavior in the unconstrained setting.

Additionally, Theorem 3.2 shows that the optimal switching point obeys a scaling law: $D - P_D^\star \sim D^\gamma$ for some exponent $\gamma$ under the FSL framework. This suggests a principled tuning strategy: one can estimate the scaling exponent via small-scale pilot experiments and extrapolate the resulting optimal switching point to large-scale training.

## 4 THE FAST CATCH-UP EFFECT: A BRIDGE TO LLM PRETRAINING

A central insight of the preceding analysis is that, for hard tasks, the optimal schedule switches to a very large batch size only in a late stage of training. We now provide a dynamical perspective that explains this phenomenon and extends naturally to LLM pretraining. We refer to this mechanism as the *fast catch-up effect*, and regard it as a key insight for practical BSS design.

**The fast catch-up effect.** Figure 1 reveals a consistent phenomenon, observed from linear regression to LLM pretraining, when the batch size is increased from small to large:

> *Once the batch size increases, the **loss rapidly collapses** onto the trajectory of large-batch training.*

In other words, although the model is trained with a small batch size for most of the trajectory, it rapidly "catches up" to the performance of training with the larger batch throughout.

To evaluate the robustness of this phenomenon in realistic large-scale settings, we conduct experiments across multiple switching times, model architectures (Dense and MoE), and scales (1.1B parameters and 1T training tokens). Figure 3 shows that fast catch-up consistently occurs in all configurations. In particular, Figure 3 (middle) presents a four-stage BSS ($640 \to 1280 \to 1920 \to 2560$). After each stage transition, the loss trajectory rapidly collapses to that of training continuously at the corresponding larger batch size. This repeated collapse across stages underscores the robustness of the fast catch-up effect.

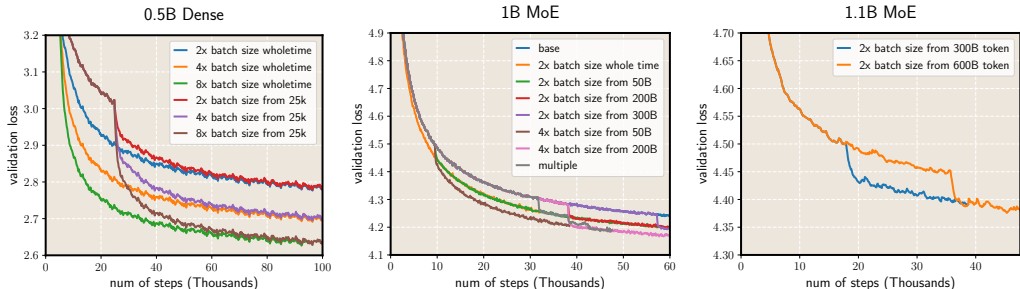

Figure 3: **The fast catch-up effect across diverse model architectures, model and data scales. Left:** A 0.5B-parameter LLaMA model trained on the C4 dataset with a base batch size of 512. **Middle:** A 1B-parameter MoE model trained on approximately 0.4T tokens with a base batch size of 640; the gray curve shows an additional 4-stage schedule beyond the two-stage runs. **Right:** A 1.1B-parameter MoE model trained on 1T tokens with a base batch size of 1024.

**The late-switch principle.** The fast catch-up effect reveals a simple yet powerful principle for batch size scheduling:

> *The validation loss of constant large-batch training can be matched by starting with a small batch and deferring the transition to the large batch until a late stage.*

Because the subsequent large-batch phase rapidly aligns with the corresponding large-batch trajectory, the final loss and total optimization steps remain unchanged. At the same time, the prolonged small-batch phase substantially reduces token consumption, lowering computational cost without sacrificing performance. We term this strategy **late switching**, and validate its effectiveness in realistic LLM pretraining in Section 5.

## 4.1 AN EXPLANATION VIA FUNCTIONAL SCALING LAWS

We now explain the fast catch-up effect using FSL. Specifically, we consider the two-stage BSS (6) and denote by $t_\star$ the switching time. Additionally, we denote by $\mathcal{E}_{B_1}(t)$ and $\mathcal{E}_{B_2}(t)$ the losses under constant batch sizes $B_1$ and $B_2$, respectively. By Theorem 2.2, for training with a constant batch size $B$ and sufficiently large $t$, the excess risk admits the decomposition $\mathcal{E}_B(t) \approx t^{-s} + \eta\sigma^2/B$, where the first term corresponds to signal learning and the second term captures noise accumulation.

**Loss gap at the switching point.** At the switching point $t_\star$, the loss gap is given by

$$G_\star := \mathcal{E}_{B_1}(t_\star) - \mathcal{E}_{B_2}(t_\star) \approx \left(t_\star^{-s} + \frac{\eta\sigma^2}{B_1}\right) - \left(t_\star^{-s} + \frac{\eta\sigma^2}{B_2}\right) = \eta\sigma^2\left(\frac{1}{B_1} - \frac{1}{B_2}\right).$$

Thus, the loss gap at the switching point arises purely from the difference in noise accumulation. The signal-learning term is identical across the two runs, since both have undergone the same optimization time $t_\star$.

**Post-switch gap decay (catch-up dynamics).** After switching to the larger batch size, FSL (4) implies that after an additional interval $\delta$, the gap decays as

$$
\begin{aligned}
\mathcal{E}_{B_1 \to B_2}(t_\star + \delta) - \mathcal{E}_{B_2}(t_\star + \delta) &= \int_0^{t_\star+\delta} \frac{\mathcal{K}(t_\star + \delta - t)}{b_{B_1 \to B_2}(t)}\, dt - \int_0^{t_\star+\delta} \frac{\mathcal{K}(t_\star + \delta - t)}{b_{B_2}(t)}\, dt \\
&= \left(\frac{\eta\sigma^2}{B_1} - \frac{\eta\sigma^2}{B_2}\right)\int_0^{t_\star} \mathcal{K}(t_\star + \delta - t)\, dt \approx G_\star\, \delta^{-(1-1/\beta)}.
\end{aligned}
\tag{7}
$$

This indicates that the catch-up dynamics progressively forget the noise accumulated during the initial small-batch phase. Importantly, the forgetting exponent depends only on the capacity exponent $\beta$ and is independent of the task difficulty $s$.

**When is the catch-up fast?** We quantify "fast" catch-up through a comparison of time scales. Define the catch-up time $\delta_\epsilon$ as the smallest $\delta$ such that

$$\mathcal{E}_{B_1 \to B_2}(t_\star + \delta) \leq (1 + \epsilon)\,\mathcal{E}_{B_2}(t_\star + \delta),$$

i.e., the switched trajectory lies within a $(1 + \epsilon)$ factor of the large-batch baseline. Combining the gap decay (7) with $\mathcal{E}_{B_2}(t_\star + \delta) \asymp (t_\star + \delta)^{-s} + \eta\sigma^2/B_2$ yields

$$\delta_\epsilon \asymp G_\star^{\frac{1}{1-1/\beta}} t_\star^{\frac{s}{1-1/\beta}} \asymp \left(\eta\sigma^2 \left(\frac{1}{B_1} - \frac{1}{B_2}\right)\right)^{\frac{1}{1-1/\beta}} t_\star^{\frac{s}{1-1/\beta}}.$$

In contrast, the large-batch loss evolves on the time scale $\delta \asymp t_\star$, since the signal term $(t_\star + \delta)^{-s}$ changes appreciably only when $\delta \gtrsim t_\star$. For hard tasks with $s < 1 - 1/\beta$, we have $\delta_\epsilon \ll t_\star$, which establishes a clear *time-scale separation*: the switched trajectory relaxes on the fast scale $\delta_\epsilon$, whereas the large-batch baseline evolves on the slow scale $t_\star$. The fast catch-up phenomenon is therefore a direct consequence of this separation of time scales. Moreover, since $\delta_\epsilon$ increases with $s$, harder tasks with smaller $s$ exhibit faster catch-up. Similarly, for a fixed switching ratio $B_2/B_1$, $(1/B_1 - 1/B_2)$ scales as $1/B_1$, so a larger base batch size $B_1$ also leads to a shorter catch-up time.

## 5 Validating Late Switching in LLM Pretraining

We now examine how the preceding theoretical results manifest in practical LLM pretraining. The experimental setup is summarized below; further details are provided in Appendix B.1.

- **Small-scale.** For small-scale experiments, we adopt the popular **NanoGPT** codebase (Karpathy, 2022) and evaluate standard dense **LLaMA** architectures (Touvron et al., 2023) on the C4 dataset (Raffel et al., 2020). Following Chinchilla law (Hoffmann et al., 2022), the total number of training tokens is set to be approximately $20\times$ the number of model parameters, a convention commonly adopted in small-scale training studies. Concretely, we consider model sizes of **50M**, **200M**, and **492M ($\approx$ 0.5B)** parameters.

- **Large-scale.** We conducted large-scale experiments using the widely adopted **Megatron-LM** codebase (Shoeybi et al., 2019). Our models are based on a sparse Mixture-of-Experts (MoE) architecture, specifically the shortcut-connected MoE proposed by Cai et al. (2025). To better reflect real-world LLM pretraining, we train our models with token-to-parameter ratios that substantially exceed the canonical 20:1 guideline, placing our experiments in a beyond-Chinchilla-optimal regime (Sardana et al., 2024). We consider two model configurations: (i) **1001M ($\approx$ 1B)** total parameters with 209M parameters activated per token, trained on approximately **0.4T** tokens; (ii) **1119M ($\approx$ 1.1B)** total parameters with 291M parameters activated per token, trained on approximately **1T** tokens.

**Fine-grained analysis of the switching time.** Figure 4 (left) shows how the final-step loss varies with the switching time. The optimal switching point occurs at approximately $70\%$ of the total training tokens, corroborating the theoretical prediction in Section 3.2: *late switching* yields improved performance.

We next validate Theorem 3.2, which establishes a power-law relation between the optimal switching point $P_D^\star$ and the total data size $D$: $D - P_D^\star \sim cD^\gamma$, for some $c > 0$ and $\gamma \in (0, 1)$. Taking logarithms yields the linear relation $\log(D - P_D^\star) = \gamma \log D + \log c$. We conduct experiments with a 50M-parameter model trained on C4, with token budgets ranging from 1.3B to 5B. For each $D$, we perform a grid search to determine the optimal switching point $P_D^\star$, and fit $\log(D - P_D^\star)$ against $\log D$ using least squares. As shown in Figure 4 (right), the fitted line indeed closely follows a power-law relation.

**Larger-scale validation of late-switch superiority.** We now turn to large-scale settings and demonstrate that late switching consistently outperforms early switching. The main results are presented in Figure 5, with additional details provided in Figure 6. Across different switching ratios, model architectures, and training scales, late switching yields consistently better performance than early switching. We further evaluate the late-switch principle in multi-stage batch-size schedules (Appendix B.5, Figures 7 and 8), where deferring batch-size increases continues to outperform early switching. Together, these results confirm that late switching is robust across model and data scales.

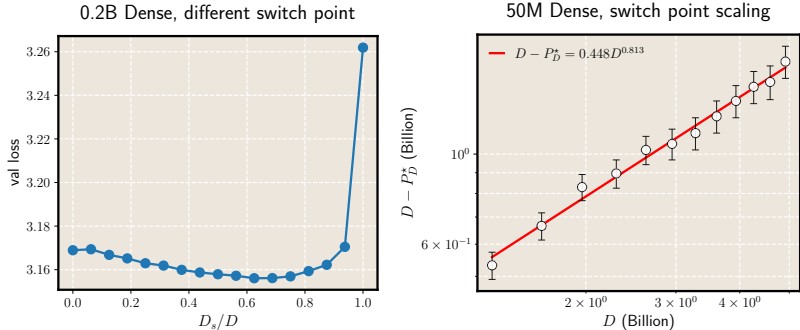

Figure 4: **Left:** Validation loss under different batch size switching points. The $x$-axis denotes the fraction of data processed before switching. **Right:** Power-law scaling between $D - P_D^\star$ and $D$. A linear fit in log–log coordinates yields $R^2 = 0.990$, supporting the predicted power-law relation.

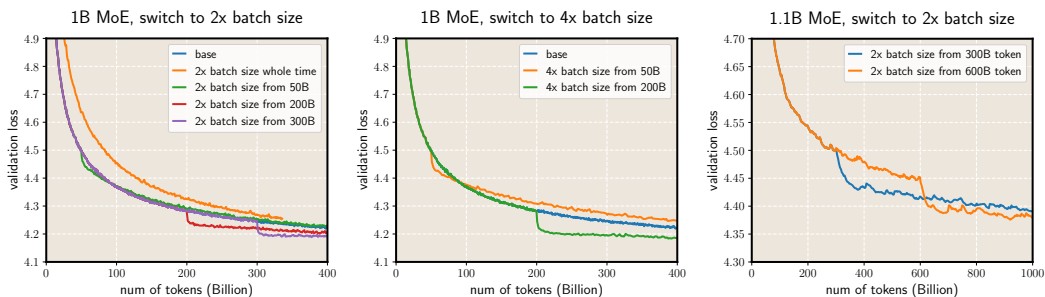

Figure 5: **Left:** Validation loss versus training tokens under different switch points for a 1B MoE model trained on 0.4T tokens; batch size increases from 640 to 1280. **Middle:** Same 1B MoE model and dataset; batch size increases from 640 to 2560. **Right:** 1.1B MoE model trained on 1T tokens; batch size increases from 1024 to 2048.

## 6 CONCLUSION

In this work, we demonstrate that the functional scaling law (FSL) provides a principled framework for analyzing batch size scheduling. We characterize the optimal batch size schedules in both the unconstrained and stage-wise settings, and show that the optimal structure depends sharply on task difficulty. In particular, hard tasks favor *late switching*: using a small batch size for most of training and transitioning to a large batch only in a late stage. To explain this structure, we uncover the *fast catch-up effect* and show that it extends beyond the theoretical setting to realistic LLM pretraining.

Several important directions remain for future work. First, the FSL framework is derived under standard SGD, whereas modern LLM training predominantly relies on adaptive optimizers. Extending the analysis to adaptive methods is therefore an important open problem.

Second, for analytical clarity, our analysis focuses on constant learning rates. This assumption is meaningful in its own right, as widely used learning rate schedules such as warmup–stable–decay maintains a constant learning rate throughout most of training (Zhai et al., 2022; Hu et al., 2024; Hägele et al., 2024). Nevertheless, understanding the joint effect of learning rate decay and batch-size scheduling—particularly how learning rate decay influences the fast catch-up effect and the resulting late-switch strategy—remains an important direction. As a preliminary exploration, we provide experiments with cosine learning-rate decay in Appendix B.6, which suggest that the fast catch-up effect continues to hold approximately. A systematic treatment of these interactions is left for future work.

ACKNOWLEDGMENT

Lei Wu is supported by the National Natural Science Foundation of China (NSFC12522120, NSFC92470122, and NSFC12288101). Binghui Li is supported by the Elite Ph.D. Program in Applied Mathematics at Peking University. Mingze Wang is supported by Young Scientists (PhD) Fund of the National Natural Science Foundation of China (No. 124B2028). We also thank Weinan E, Zilin Wang, Shaowen Wang, Kairong Luo, Haodong Wen and Kaifeng Lyu for many helpful discussions, and the anonymous reviewers for their valuable feedback.

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

# Appendix

## A  THEORETICAL SETUP AND PROOFS

### A.1  INTERPRETATION OF THE SOURCE AND CAPACITY CONDITIONS

In this section, we provide a detailed description of the parameters of feature-space linear regression (aka power-law kernel regression in Li et al. (2025a)) and their interpretation in the context of LLM pretraining. Let $\widehat{\phi}_j := \phi_j / \lambda_j^{1/2}$ for $j \in [N]$, so that $\{\widehat{\phi}_j\}_{j=1}^N$ forms an orthonormal basis of $L^2(\mathcal{D})$.

**Model Capacity $\beta$:**  A model of the form

$$f(\cdot; \boldsymbol{\theta}) = \sum_{j=1}^N \theta_j \phi_j = \sum_{j=1}^N \theta_j \lambda_j^{1/2} \widehat{\phi}_j \approx \sum_{j=1}^N \theta_j \, j^{-\beta/2} \widehat{\phi}_j$$

shows that higher-index features are increasingly down-weighted by the factor $j^{-\beta/2}$. As $\beta$ increases, the spectrum decays more rapidly, causing the model to **effectively** rely on fewer features.

Additionally, for a fixed target function $f^\star$, one can use different (potentially nonlinear) feature maps $\phi$ (and consequently, different values of $\beta$). The value of $\beta$ reflects the **capacity** of the chosen features. For instance, consider $\phi(\boldsymbol{x}) = \nabla_\theta \mathcal{N}(\boldsymbol{x}; \theta)$, where $\mathcal{N}(\cdot; \theta)$ denotes a neural network. In this case, $\phi(\boldsymbol{x})$ corresponds to neural tangent features, and the associated kernel

$$K_\phi(\boldsymbol{x}, \boldsymbol{x}') := \phi(\boldsymbol{x})^\top \phi(\boldsymbol{x}')$$

is known as the neural tangent kernel (NTK). Here, the network depth and activation functions govern the spectral decay, determining the effective exponent $\beta$.

**Task Difficulty $s$:**  The target function admits the expansion

$$f^\star = \sum_{j=1}^N \theta_j^\star \phi_j \approx \sum_{j=1}^N j^{-1/2} \lambda_j^{s/2} \widehat{\phi}_j \approx \sum_{j=1}^N j^{-(s\beta+1)/2} \widehat{\phi}_j.$$

Since $\{\widehat{\phi}_j\}$ are orthonormal, this assumption implies that the spectral energy of $f^\star$ decays according to a power law. The exponent $\alpha := s\beta$ thus quantifies the task's **intrinsic difficulty**, which depends only on the target function itself and is independent of the model's spectrum. In contrast, $s$ measures the **relative difficulty** with respect to a model of capacity $\beta$: for a fixed $f^\star$ (and fixed $\alpha$), adopting a higher-capacity model (smaller $\beta$) increases $s = \alpha/\beta$, making the task relatively easier. In other words, the same task appears easier to a higher-capacity model.

**Connection with LLM Pretraining.**  In the context of large language model (LLM) pretraining, the parameter $\beta$ reflects the **model architecture** and determines its capacity. Specifically, $\beta$ is influenced by factors such as the depth of the model, the activation functions, and the choice of feature map. A model with a larger capacity (smaller $\beta$) has a spectrum that decays more slowly, allowing it to utilize a broader range of features, whereas a model with a smaller capacity (larger $\beta$) down-weights higher-index features more rapidly. On the other hand, the parameter $s$ reflects the **difficulty of the task** relative to the model architecture. It quantifies how challenging a particular task is for a given model capacity $\beta$. For a fixed target function $f^\star$, increasing the model's capacity (reducing $\beta$) leads to a larger value of $s$, making the task easier. In other words, the same task will appear **easier** to a model with a higher capacity, because the model can better accommodate the complexity of the task due to its architecture.

### A.2  PROOF OF THEOREM 2.2 (SELF-CONTAINED DERIVATION OF FSL)

A key insight that makes the above SDE (3) analytically tractable is the anisotropic noise structure, which can be formalized as follows:

**Lemma A.1** (Anisotropic noise). *For any $\boldsymbol{\theta} \in \mathbb{R}^N$, it holds that*

$$\left(2\mathcal{E}(\boldsymbol{\theta}) + \sigma^2\right) \mathbf{H} \preceq \boldsymbol{\Sigma}(\boldsymbol{\theta}) \preceq \left(4\mathcal{E}(\boldsymbol{\theta}) + \sigma^2\right) \mathbf{H}.$$

Lemma A.1 demonstrates that the noise covariance $\boldsymbol{\Sigma}(\boldsymbol{\theta})$ approximately admits a closed-form expression: $\boldsymbol{\Sigma}(\boldsymbol{\theta}) \propto \mathcal{R}(\boldsymbol{\theta})\mathbf{H}$, as observed in (Mori et al., 2022; Wu et al., 2022b; Wang & Wu, 2023). This closed-form expression enables a precise characterization of the noise dynamics, thus providing a framework for tracking the SGD training dynamics.

*Proof.* For a given data point $\boldsymbol{z} = (\boldsymbol{x}, y)$, we define the point-wise risk as $\ell(\boldsymbol{z}; \boldsymbol{\theta}) := \frac{1}{2}(\boldsymbol{\theta}^\top \boldsymbol{\phi}(\boldsymbol{x}) - y)^2$. By definition of $\ell(\boldsymbol{z}; \boldsymbol{\theta})$ and $\mathcal{R}(\boldsymbol{\theta})$, we have

$$\nabla \ell(\boldsymbol{z}; \boldsymbol{\theta}) = \boldsymbol{\phi}(\boldsymbol{x})\boldsymbol{\phi}(\boldsymbol{x})^\top (\boldsymbol{\theta} - \boldsymbol{\theta}^\star) - \boldsymbol{\phi}(\boldsymbol{x})\epsilon,$$

$$\nabla \mathcal{R}(\boldsymbol{\theta}) = \mathbb{E}[\nabla \ell(\boldsymbol{z}; \boldsymbol{\theta})] = \mathbf{H}(\boldsymbol{\theta} - \boldsymbol{\theta}^\star).$$

For the stochastic mini-batch gradient noise $\boldsymbol{\xi} := \nabla \ell(\boldsymbol{z}; \boldsymbol{\theta}) - \nabla \mathcal{R}(\boldsymbol{\theta})$, we have

$$\boldsymbol{\xi} = \boldsymbol{\phi}(\boldsymbol{x})\boldsymbol{\phi}(\boldsymbol{x})^\top (\boldsymbol{\theta} - \boldsymbol{\theta}^\star) - \boldsymbol{\phi}(\boldsymbol{x})\epsilon - \mathbf{H}(\boldsymbol{\theta} - \boldsymbol{\theta}^\star).$$

Hence, the covariance matrix $\boldsymbol{\Sigma}(\boldsymbol{\theta}) = \mathbb{E}[\boldsymbol{\xi}\boldsymbol{\xi}^\top | \boldsymbol{\theta}]$ satisfies

$$\boldsymbol{\Sigma}(\boldsymbol{\theta}) = \left(\mathbb{E}\left[\boldsymbol{\phi}(\boldsymbol{x})\boldsymbol{\phi}(\boldsymbol{x})^\top \boldsymbol{u}\boldsymbol{u}^\top \boldsymbol{\phi}(\boldsymbol{x})\boldsymbol{\phi}(\boldsymbol{x})^\top\right] - \mathbf{H}\boldsymbol{u}\boldsymbol{u}^\top\mathbf{H}\right) + \sigma^2\mathbf{H},$$

where $\boldsymbol{u} = \boldsymbol{\theta} - \boldsymbol{\theta}^\star$. Let $\boldsymbol{M} := \mathbb{E}\left[\boldsymbol{\phi}(\boldsymbol{x})\boldsymbol{\phi}(\boldsymbol{x})^\top \boldsymbol{u}\boldsymbol{u}^\top \boldsymbol{\phi}(\boldsymbol{x})\boldsymbol{\phi}(\boldsymbol{x})^\top\right]$ and $\boldsymbol{M}_{ij}$ be $(i,j)$ entry of $\boldsymbol{M}$. Calculating $\boldsymbol{M}_{ij}$ using Wick's probability theorem

$$\boldsymbol{M}_{ij} = \sum_{k,l} \boldsymbol{u}_k\boldsymbol{u}_l\mathbb{E}[\phi_i(\boldsymbol{x})\phi_k(\boldsymbol{x})\phi_l(\boldsymbol{x})\phi_j(\boldsymbol{x})] = \sum_{k,l} \boldsymbol{u}_k\boldsymbol{u}_l(\mathbf{H}_{ik}\mathbf{H}_{lj} + \mathbf{H}_{il}\mathbf{H}_{kj} + \mathbf{H}_{ij}\mathbf{H}_{kl}).$$

Recognizing each term, we know

$$\sum_{k,l} \boldsymbol{u}_k\boldsymbol{u}_l\mathbf{H}_{ik}\mathbf{H}_{lj} = (\mathbf{H}\boldsymbol{u}\boldsymbol{u}^\top\mathbf{H})_{ij}$$

$$\sum_{k,l} \boldsymbol{u}_k\boldsymbol{u}_l\mathbf{H}_{il}\mathbf{H}_{kj} = (\mathbf{H}\boldsymbol{u}\boldsymbol{u}^\top\mathbf{H})_{ij}$$

$$\sum_{k,l} \boldsymbol{u}_k\boldsymbol{u}_l\mathbf{H}_{kl}\mathbf{H}_{ij} = (\boldsymbol{u}^\top\mathbf{H}\boldsymbol{u})\mathbf{H}_{ij}.$$

Hence

$$\boldsymbol{M} = 2\mathbf{H}\boldsymbol{u}\boldsymbol{u}^\top\mathbf{H} + (\boldsymbol{u}^\top\mathbf{H}\boldsymbol{u})\mathbf{H}$$

$$\boldsymbol{\Sigma}(\boldsymbol{\theta}) = \mathbf{H}\boldsymbol{u}\boldsymbol{u}^\top\mathbf{H} + (\boldsymbol{u}^\top\mathbf{H}\boldsymbol{u})\mathbf{H} + \sigma^2\mathbf{H}.$$

Noting that $\boldsymbol{u}^\top\mathbf{H}\boldsymbol{u} = 2\mathcal{E}(\boldsymbol{\theta})$, for any vector $\boldsymbol{x}$ with the same shape of $\boldsymbol{u}$, we have

$$\boldsymbol{x}^\top(\mathbf{H}\boldsymbol{u}\boldsymbol{u}^\top\mathbf{H})\boldsymbol{x} = \langle \boldsymbol{u}, \boldsymbol{x}\rangle_\mathbf{H}^2 \leq \langle \boldsymbol{u}, \boldsymbol{u}\rangle_\mathbf{H}\langle \boldsymbol{x}, \boldsymbol{x}\rangle_\mathbf{H} = (\boldsymbol{u}^\top\mathbf{H}\boldsymbol{u})\boldsymbol{x}^\top\mathbf{H}\boldsymbol{x}.$$

Hence $\mathbf{H}\boldsymbol{u}\boldsymbol{u}^\top\mathbf{H} \preceq (\boldsymbol{u}^\top\mathbf{H}\boldsymbol{u})\mathbf{H}$ and

$$\boldsymbol{\Sigma}(\boldsymbol{\theta}) \succeq (\boldsymbol{u}^\top\mathbf{H}\boldsymbol{u})\mathbf{H} + \sigma^2\mathbf{H} = (2\mathcal{E}(\boldsymbol{\theta}) + \sigma^2)\mathbf{H}$$

$$\boldsymbol{\Sigma}(\boldsymbol{\theta}) \preceq 2(\boldsymbol{u}^\top\mathbf{H}\boldsymbol{u})\mathbf{H} + \sigma^2\mathbf{H} = (4\mathcal{E}(\boldsymbol{\theta}) + \sigma^2)\mathbf{H}.$$

$\square$

Now we proceed to the main theorem.

*Proof.* For the SDE (3)

$$\mathrm{d}\boldsymbol{\theta}_t = -\nabla \mathcal{R}(\boldsymbol{\theta}_t)\, \mathrm{d}t + \sqrt{\frac{\eta}{b(t)}\boldsymbol{\Sigma}(\boldsymbol{\theta}_t)}\, \mathrm{d}\boldsymbol{B}_t.$$

For each coordinate $j$, we define $p_j := \boldsymbol{e}_j^\top \boldsymbol{\Sigma}(\boldsymbol{\theta}_t)\boldsymbol{e}_j$, we have

$$\mathrm{d}\theta_j(t) = -\lambda_j(\theta_j - \theta_j^\star)\, \mathrm{d}t + \sqrt{\frac{\eta}{b(t)}p_j}\, \mathrm{d}B_j(t).$$

Applying Itô's formula to $(\theta_j - \theta_j^\star)^2$, we obtain

$$\mathbb{E}[(\theta_j - \theta_j^\star)^2] = |\theta_j^\star|^2 e^{-2\lambda_j t} + \int_0^t e^{-2\lambda_j(t-z)} \frac{\eta}{b(z)} p_j \, dz.$$

$$2\mathbb{E}[\mathcal{E}(\boldsymbol{\theta}_t)] = \sum_{j=1}^\infty \lambda_j |\theta_j^\star|^2 e^{-2\lambda_j t} + \sum_{j=1}^\infty \lambda_j \int_0^t e^{-2\lambda_j(t-z)} \frac{\eta}{b(z)} p_j \, dz.$$

By Lemma A.1, it is trivial that $p_j = \boldsymbol{e}_j^\top \boldsymbol{\Sigma}(\boldsymbol{\theta}_t)\boldsymbol{e}_j \eqsim \lambda_j(\mathcal{E}(\boldsymbol{\theta}_t) + \sigma^2/2)$, we have the following Volterra equation:

$$2\mathbb{E}[\mathcal{E}(\boldsymbol{\theta}_t)] \eqsim \sum_{j=1}^\infty \lambda_j |\theta_j^\star|^2 e^{-2\lambda_j t} + \sum_{j=1}^\infty \lambda_j \int_0^t e^{-2\lambda_j(t-z)} \frac{\eta}{b(z)} p_j \, dz$$

$$\eqsim e(t) + \int_0^t \frac{\eta}{b(z)} \mathcal{K}(t-z)(\mathbb{E}[\mathcal{E}(\boldsymbol{\theta}_z)] + \sigma^2) \, dz,$$

where

$$e(t) = \sum_{j=1}^\infty \lambda_j |\theta_j^\star|^2 e^{-2\lambda_j t} \eqsim \int_0^1 u^{s-1} e^{-2ut} \, du.$$

$$\mathcal{K}(t) = \sum_{j=1}^\infty \lambda_j^2 e^{-2\lambda_j t} \eqsim \int_0^1 u^{1-\frac{1}{\beta}} e^{-2ut} \, du.$$

Let $f(t) := \mathbb{E}[\mathcal{E}(\boldsymbol{\theta}_t)]$, $g(t) := e(t) + \sigma^2 \int_0^t \mathcal{K}(t-z)\frac{\eta}{b(z)} \, dz$, and define the linear operator

$$\mathcal{T}f(t) := \int_0^t \mathcal{K}(t-z)\frac{\eta}{b(z)} f(z) \, dz.$$

With this notation, the Volterra equation admits the compact representation $f = g + \mathcal{T}f$. Formally, the solution can be expressed via the Neumann series expansion:

$$f = (\mathcal{I} - \mathcal{T})^{-1}g = \sum_{i=0}^\infty \mathcal{T}^i g.$$

Note that $\mathcal{K} * \mathcal{K}(t) = 2\int_0^{t/2} \mathcal{K}(t-z)\mathcal{K}(z) \, dz \leq 2\mathcal{K}(t/2)\int_0^{t/2} \mathcal{K}(z) \, dz \lesssim \mathcal{K}(t/2) \lesssim \mathcal{K}(t)$, by $\eta/b(t) \leq \eta$,

$$\mathcal{T}^2 g(t) \leq \eta \int_0^t \mathcal{K} * \mathcal{K}(t-z)\frac{\eta}{b(z)} g(z) \, dz \lesssim \eta \int_0^t \mathcal{K}(t-z)\frac{\eta}{b(z)} g(z) \, dz = \eta \mathcal{T}g(t).$$

Hence, we have

$$g(t) + \mathcal{T}g(t) \leq f(t) \leq g(t) + \mathcal{T}g(t) + \sum_{k=2}^\infty \eta^{k-1}\mathcal{T}g(t) \lesssim g(t) + \frac{1}{1-\eta}\mathcal{T}g(t).$$

As a result,

$$\mathbb{E}[\mathcal{R}(\boldsymbol{\theta}_t)] - \frac{1}{2}\sigma^2 = f(t) \eqsim g(t) + \mathcal{T}g(t) \eqsim \frac{1}{t^s} + \eta \int_0^t \frac{\mathcal{K}(t-r)}{b(r)} \, dr.$$

$\square$

### A.3  PROOF OF THEOREM 3.1 (SHAPE-UNCONSTRAINED OPTIMAL BSS)

**Lemma A.2.** *Define the feasible region of BSS under data $D$:*

$$\mathcal{B}_D := \left\{ (T, b) \ \Big| \ T \in \mathbb{R}_{>0}, b \in L^1(0, T), b(t) > 0 \ a.e., \int_0^T b(t) \, \mathrm{d}t = D \right\}.$$

*Consider the following optimal batch size scheduling problem:*

$$\min_{(T,b)\in\mathcal{B}_D} \mathcal{E}[T, b] := \frac{1}{T^s} + \int_0^T \frac{\mathcal{K}(T - t)}{b(t)} \, \mathrm{d}t.$$

*The optimal batch size schedule obeys*

$$b(t) \asymp \frac{(T^\star - t + 1)^{\frac{1}{2\beta} - 1}}{(T^\star + 1)^{\frac{1}{2\beta}}} D,$$

*with*

$$T^\star \asymp D^{\frac{1}{1/\beta + s}}, \quad \mathcal{E}_D^\star \asymp D^{-\frac{s\beta}{1 + s\beta}}.$$

*Proof.* We first minimize the second term of the loss under fixed $T$. By the Cauchy-Schwarz inequality, we have

$$\left( \int_0^T \frac{\mathcal{K}(T - t)}{b(t)} \, \mathrm{d}t \right) \left( \int_0^T b(t) \, \mathrm{d}t \right) \geq \left( \int_0^T \sqrt{\mathcal{K}(T - t)} \, \mathrm{d}t \right)^2.$$

Equality holds when
$$b(t) = C\sqrt{\mathcal{K}(T - t)} \asymp C(T - t + 1)^{\frac{1}{2\beta} - 1},$$

where $C$ ensures $\int_0^T b(t) \, \mathrm{d}t = D$. The minimizer must satisfy the above equality, combining with $\int_0^T b(t) \, \mathrm{d}t = D$, we have

$$b(t) \asymp D \frac{(T - t + 1)^{\frac{1}{2\beta} - 1}}{(T + 1)^{\frac{1}{2\beta}}},$$

and consequently,

$$\int_0^T \frac{\mathcal{K}(T - t)}{b(t)} \, \mathrm{d}t = \frac{\left( \int_0^T \mathcal{K}^{1/2}(T - t) \, \mathrm{d}t \right)^2}{\int_0^T b(t) \, \mathrm{d}t} \asymp \frac{(T + 1)^{1/\beta}}{D}.$$

Consequently, define

$$g(T) := \min_{(T,b)\in\mathcal{B}_D} \frac{1}{T^s} + \int_0^T \frac{\mathcal{K}(T - t)}{b(t)} \, \mathrm{d}t \asymp \frac{1}{T^s} + \frac{(T + 1)^{1/\beta}}{D}.$$

Minimizing the above risk with respect to T, we obtain the optimal $T^\star$

$$T^\star \asymp D^{\frac{1}{1/\beta + s}}.$$

Substituting $T^\star$ back, the minimum $\mathcal{E}$ satisfies

$$\mathcal{E}_D^\star = (T^\star)^{-s} + \frac{(T^\star)^{1/\beta}}{D} \asymp D^{-\frac{s}{1/\beta + s}} = D^{-\frac{s\beta}{1 + s\beta}}.$$

The corresponding optimal batch size schedule satisfies

$$b^\star(t) \asymp \frac{(T^\star - t + 1)^{\frac{1}{2\beta} - 1}}{(T^\star + 1)^{\frac{1}{2\beta}}} D.$$

$\square$

**Lemma A.3.** *Define the feasible region of BSS under data $D$:*

$$\mathcal{B}_D := \left\{ (T, b) \ \middle|\ T \in \mathbb{R}_{>0}, b \in L^1(0, T), B_1 \leq b(t) \leq B_2 \ a.e., \int_0^T b(t)\, \mathrm{d}t = D \right\}.$$

*Consider the following optimal batch size scheduling problem:*

$$\min_{(T, b) \in \mathcal{B}_D} \mathcal{E}[T, b] := \frac{1}{T^s} + \int_0^T \frac{\mathcal{K}(T - t)}{b(t)}\, \mathrm{d}t,$$

*The optimal batch size schedule must take one of two possible forms:*
*(i)*

$$b^\star(t) = C_1 \sqrt{\mathcal{K}(T - t)} \asymp C_2(T - t + 1)^{\frac{1}{2\beta} - 1} \ for\ 0 \leq t \leq T,$$

*with $b^\star(0) \geq B_1$.*
*(ii)*

$$b^\star(t) = \begin{cases} B_1, & for\ t < T_1, \\ C_1 \sqrt{\mathcal{K}(T - t)} \asymp C_2(T - t + 1)^{\frac{1}{2\beta} - 1}, & for\ t \geq T_1, \end{cases}$$

*where $T_1$ is determined by the boundary-matching condition $C_2(T - T_1 + 1)^{1/(2\beta) - 1} = B_1$.*

*Proof.* We now consider the constrained problem under fixed $T$ with $B_1 \leq b(t) \leq B_2$. Since the integrand $\mathcal{K}(T - t)/b(t)$ is convex for $b(t) > 0$, and the constraints are linear, so Slater's condition holds. Consequently, any point satisfying the KKT conditions is a global minimizer. Consider the Lagrangian

$$L[T, b] := \int_0^T \left( \frac{\mathcal{K}(T - t)}{b(t)} + \lambda b(t) + \mu(t)(B_1 - b(t)) + \xi(t)(b(t) - B_2) \right) \mathrm{d}t - \lambda D$$

with $\mu(t) \geq 0$ and $\xi(t) \geq 0$. The stationarity condition is given by

$$-\frac{\mathcal{K}(T - t)}{b(t)^2} + \lambda - \mu(t) + \xi(t) = 0. \tag{8}$$

The complementary slackness conditions are

$$\mu(t)(B_1 - b(t)) = 0, \ \xi(t)(b(t) - B_2) = 0, \ B_1 \leq b(t) \leq B_2, \ \mu(t) \geq 0, \ \xi(t) \geq 0. \tag{9}$$

Define

$$\mathcal{A} := \{t | B_1 < b(t) < B_2\}, \ \mathcal{I}_1 := \{t | b(t) = B_1\}, \ \mathcal{I}_2 := \{t | b(t) = B_2\}.$$

In $\mathcal{A}$, both constraints are inactive, thus $\mu(t) = \xi(t) = 0$. The stationarity condition (8) yields

$$-\frac{\mathcal{K}(T - t)}{b(t)^2} + \lambda = 0.$$

Solving for $b(t)$, we obtain

$$b(t) = \sqrt{\frac{\mathcal{K}(T - t)}{\lambda}}.$$

In $\mathcal{I}_1$, we have $b(t) = B_1$ and $\xi(t) = 0$; In $\mathcal{I}_2$, we have $b(t) = B_2$, $\mu(t) = 0$. By stationarity (8) and complementary slackness(9), we have the following two relations on $\mathcal{I}_1$ and $\mathcal{I}_2$: For $\mathcal{I}_1$, $-\frac{\mathcal{K}(T-t)}{B_1^2} + \lambda - \mu(t) = 0$ implies $\mu(t) = \lambda - \frac{\mathcal{K}(T-t)}{B_1^2} \geq 0$, which further yields $b(t) = B_1 \geq \sqrt{\frac{\mathcal{K}(T-t)}{\lambda}}$; For $\mathcal{I}_2$, $-\frac{\mathcal{K}(T-t)}{B_2^2} + \lambda + \xi(t) = 0$ implies $\xi(t) = \frac{\mathcal{K}(T-t)}{B_2^2} - \lambda \geq 0$, which in turn yields $b(t) = B_2 \leq \sqrt{\frac{\mathcal{K}(T-t)}{\lambda}}$. Therefore, the optimal batch size schedule is given by

$$b^\star(t) = \mathrm{clip}\left( \sqrt{\frac{\mathcal{K}(T - t)}{\lambda}}, B_1, B_2 \right) = \mathrm{clip}(C \sqrt{\mathcal{K}(T - t)}, B_1, B_2),$$

where $\mathrm{clip}(x, a, b) = \max\{a, \min\{x, b\}\}$. Exploiting the monotonicity of $\mathcal{K}(T - t)$, the schedule admits the following piecewise form:

$$b^\star(t) = \begin{cases} B_1, & for\ t < T_1, \\ C \sqrt{\mathcal{K}(T - t)} \asymp C'(T - t)^{\frac{1}{2\beta} - 1}, & for\ T_1 \leq t \leq T_2, \\ B_2, & for\ t > T_2, \end{cases}$$

where $C$ and $C'$ are problem-dependent constants that depend on $D, T, \beta, s, B_1, B_2$.

(i) When $T_1 = 0$, the schedule takes the **first** form

$$b^\star(t) = C_1\sqrt{\mathcal{K}(T-t)} \asymp C_2(T-t+1)^{\frac{1}{2\beta}-1} \text{ for } 0 \leq t \leq T,$$

with $b^\star(0) \geq B_1$.

(ii) When $T_1 > 0$, the schedule takes the **second** form

$$b^\star(t) = \begin{cases} B_1, & \text{for } t < T_1 \\ C_1\sqrt{\mathcal{K}(T-t)} \asymp C_2(T-t+1)^{\frac{1}{2\beta}-1}, & \text{for } t \geq T_1, \end{cases}$$

where $T_1$ is determined by the boundary-matching condition $C_2(T-T_1+1)^{1/(2\beta)-1} = B_1$. $\qquad\square$

Now we proceed to the main theorem. For the main theorem, we only consider Lemma A.3 with $B_1 = B_{\min}$ and $B_2 = \infty$.

*Proof.* (I) We now consider whether the optimal batch size schedule $b^\star(t)$ in Lemma A.2 can satisfy the constraint $b(t) \geq B_1$ under the easy-task regime. Since $b^\star(t)$ is non-decreasing, and

$$b^\star(0) \asymp D/(T^\star + 1) \asymp D^{1-\frac{1}{1/\beta+s}} \gtrsim 1.$$

It follows that under the easy task regime, the constraint $b^\star(t) \geq B_1$ is automatically satisfied when $D$ is sufficiently large. Consequently, we have

$$T^\star \asymp D^{\frac{\beta}{1+s\beta}}, \quad \mathcal{E}_D^\star \asymp D^{-\frac{s\beta}{1+s\beta}}.$$

We have $b^\star(t) \asymp C_2(T^\star - t + 1)^{\frac{1}{2\beta}-1}$, where the constant $C_2$ is determined from the budget constraint

$$\int_0^{T^\star} b^\star(t)\, \mathrm{d}t = C_2 \int_0^{T^\star} (t+1)^{1/(2\beta)-1}\, \mathrm{d}t = D,$$

Solving for $C_2$, we obtain

$$C_2 \asymp D(T^\star)^{-1/(2\beta)} = DD^{\frac{-1/2}{1+s\beta}} = D^{\frac{1/2+s\beta}{1+s\beta}}.$$

(II) Under the hard task regime with $s < 1 - 1/\beta$ ($s = 1 - 1/\beta$ is trivially similar), the unconstrained solution in Lemma A.2 is infeasible due to $T^\star \gtrsim D$, which trivially violates the constraint. We therefore analyze the constrained candidates in Lemma A.3 and determine which achieves a lower objective value. We first consider the **second** form in Lemma A.3.

$$b^\star(t) = \begin{cases} B_1, & \text{for } t < T_1 \\ B_1\left(\frac{T_1+T_2-t+1}{T_2+1}\right)^{\frac{1}{2\beta}-1}, & \text{for } T_1 \leq t \leq T := T_1 + T_2 \end{cases} \tag{10}$$

The data-budget constraint implies

$$T_1 = \frac{D}{B_1} - \int_0^{T_2} \left(\frac{t+1}{T_2+1}\right)^{\frac{1}{2\beta}-1}\, \mathrm{d}t \tag{11}$$

Let $a = 1/(2\beta) - 1$. We consider the objective function

$$\mathcal{E} := \frac{1}{T^s} + \int_0^T \frac{\mathcal{K}(T-t)}{b(t)}\, \mathrm{d}t$$

Substituting (10) into the above objective yields

$$\mathcal{E} = T^{-s} + \frac{1}{B_1}\int_{T_2}^T (t+1)^{2a}\, \mathrm{d}t + \frac{1}{B_1}(T_2+1)^a \int_0^{T_2} (t+1)^a\, \mathrm{d}t$$

Let these three parts be $\mathcal{E}_1$, $\mathcal{E}_2$, and $\mathcal{E}_3$, respectively. Their derivatives with respect to $T_2$ are

$$\frac{\mathrm{d}\mathcal{E}_1}{\mathrm{d}T_2} = -sT^{-s-1}\frac{\mathrm{d}T}{\mathrm{d}T_2}$$

$$\frac{\mathrm{d}\mathcal{E}_2}{\mathrm{d}T_2} = (T+1)^{2a}\frac{\mathrm{d}T}{\mathrm{d}T_2} - (T_2+1)^{2a}$$

$$\frac{\mathrm{d}\mathcal{E}_3}{\mathrm{d}T_2} = \frac{2a+1}{a+1}(T_2+1)^{2a} - \frac{a}{a+1}(T_2+1)^{a-1}$$

The optimal $T_2$ must satisfy

$$\frac{\mathrm{d}\mathcal{E}}{\mathrm{d}T_2} = \frac{\mathrm{d}\mathcal{E}_1}{\mathrm{d}T_2} + \frac{\mathrm{d}\mathcal{E}_2}{\mathrm{d}T_2} + \frac{\mathrm{d}\mathcal{E}_3}{\mathrm{d}T_2} = 0$$

For the regime $D \gtrsim 1$, by Equation (11), we have $T_1 = D/B_1 - 2\beta\left[(T_2+1) - (T_2+1)^{1-\frac{1}{2\beta}}\right]$, this implies $T \gtrsim 1$. Moreover, we must have $T_2 \gtrsim 1$; otherwise, the expression above would be dominated by its first term and become negative when $D \gtrsim 1$. Keeping only the dominant terms gives

$$T^{-s-1} \approx (T_2+1)^{2a} \approx T_2^{1/\beta-2}.$$

Hence,

$$T_2 \approx T^{\frac{s\beta+\beta}{2\beta-1}}.$$

Since $T \lesssim D$, it follows that $T_2 \lesssim D^{\frac{s\beta+\beta}{2\beta-1}}$ and therefore

$$\int_0^{T_2} \left(\frac{t+1}{T_2+1}\right)^{\frac{1}{2\beta}-1} \mathrm{d}t = 2\beta\left((T_2+1) - (T_2+1)^{1-\frac{1}{2\beta}}\right) \lesssim D^{\frac{s\beta+\beta}{2\beta-1}}.$$

By the hard-task regime condition, $s\beta + \beta < 2\beta - 1$. Together with (11), this yields $T_1 \approx D$, hence, $T \approx D$, which yields

$$T_2 \approx D^{\frac{s\beta+\beta}{2\beta-1}} \text{ and } \mathcal{E} \approx D^{-s}.$$

In particular, $\mathcal{E}$ is now dominated by the signal learning term with $B_1 T_1 \geq (1-\epsilon)D$. We next consider the **first** form in Lemma A.3.

$$b(t) = C_1\sqrt{\mathcal{K}(T-t)} \approx C_2(T-t+1)^{1/(2\beta)-1} \text{ for } 0 \leq t \leq T.$$

Since $b(0) \geq B_1$, we have

$$D = \int_0^T b(t)\,\mathrm{d}t \geq B_1\int_0^T \left(\frac{t+1}{T+1}\right)^{1/(2\beta)-1}\mathrm{d}t$$
$$= 2\beta B_1\left((T+1) - (T+1)^{1-\frac{1}{2\beta}}\right) \geq (2\beta-\epsilon)TB_1,$$

which implies the intrinsic term satisfies

$$T \leq \frac{D}{(2\beta-\epsilon)B_1}.$$

However, in the **second** form, $\mathcal{E}$ is dominated by the signal learning term and satisfies $T \geq (1-\epsilon)D/B_1$. Therefore, the signal-learning term under the first form is worse than that under the second form by at least a constant factor. Since $\mathcal{E}$ in the second form is signal-dominated, this constant-factor improvement carries over to the total error, implying that the second form strictly dominates the first and is therefore optimal. Finally, from

$$C_2(T_2+1)^{\frac{1}{2\beta}-1} = B_1,$$

we obtain $C_2 \approx D^{\frac{s+1}{2}}$, which gives the desired scaling for $C_2$. $\qquad\square$

### A.4 PROOF OF THEOREM 3.2 (OPTIMAL TWO-STAGE BSS)

*Proof.* Recalling that

$$b^P_{B_1 \to B_2}(t) = \begin{cases} B_1, & 0 < t \leq T_{s,P}, \\ B_2, & T_{s,P} < t < T_P, \end{cases} \qquad T_{s,P} = \frac{P}{B_1}, \quad T_P = \frac{P}{B_1} + \frac{D-P}{B_2}.$$

For clarity, we omit the explicit dependence on $D$ in $P(D)$. Following Theorem 2.2,

$$\frac{\mathrm{d}}{\mathrm{d}P}\mathcal{E}_{B_1 \to B_2}(T_P) = \left(\frac{1}{B_1} - \frac{1}{B_2}\right)\left[-sT_P^{-s-1} + \left(\frac{\mathcal{K}(T_P)}{B_1} + \frac{\mathcal{K}((D-P)/B_2)}{B_2}\right)\right].$$

Since $B_1 < B_2$, we have $1/B_1 - 1/B_2 > 0$. Note that under the two-stage batch schedule setting,

$$T_P = \frac{P}{B_1} + \frac{D-P}{B_2}, \quad \frac{\mathrm{d}T_P}{\mathrm{d}P} = \frac{1}{B_1} - \frac{1}{B_2} > 0.$$

Since $T_P \approx D$,

$$-sT_P^{-s-1} \approx -D^{-s-1}, \quad \frac{\mathcal{K}(T_P)}{B_1} \approx \frac{D^{-(2-\frac{1}{\beta})}}{B_1}.$$

(I) Under the hard-task regime with $s < 1 - 1/\beta$ ($s = 1 - 1/\beta$ is trivially similar), since $D^{-(2-1/\beta)} = o(D^{-s-1})$, the minimizer $P^\star$ must satisfy the stationary point condition:

$$\frac{\mathrm{d}}{\mathrm{d}P}\mathcal{E}_{B_1 \to B_2}(T_P)\Big|_{P=P^\star} = 0.$$

In particular,

$$\mathcal{K}\left(\frac{D - P^\star}{B_2}\right) \approx D^{-s-1}.$$

Since $\mathcal{K}((D - P^\star)/B_2) \to 0$, By the monotonicity of $\mathcal{K}$, this implies $D - P^\star \to \infty$. We have

$$(D - P^\star)^{-(2-1/\beta)} \approx D^{-s-1},$$

$$D - P^\star \approx D^{\frac{s+1}{2-1/\beta}}.$$

We now verify that the stationary point $P^\star$ is a minimizer and corresponding $T_{P^\star}$:

$$\frac{\mathrm{d}^2}{\mathrm{d}P^2}\mathcal{E}_{B_1 \to B_2}(T_P) = \left(\frac{1}{B_1} - \frac{1}{B_2}\right)\left[s(s+1)T_P^{-s-2}T_P' + \frac{\mathcal{K}'(T_P)}{B_1}T_P' - \frac{\mathcal{K}'((D-P)/B_2)}{B_2^2}\right].$$

Note that $T_P' > 0$ and $\mathcal{K}' < 0$, it suffices to prove

$$\left(\frac{\mathcal{K}'(T_P)}{B_1}T_P' - \frac{\mathcal{K}'((D-P)/B_2)}{B_2^2}\right)\Big|_{P=P^\star} > 0.$$

The above inequality is trivial since

$$(D - P^\star)/B_2 = o(T_{P^\star}) \Rightarrow -\mathcal{K}'(T_{P^\star}) = o(-\mathcal{K}'((D-P^\star)/B_2)).$$

(II) Under the easy-task regime, since $D^{-s-1} = o(D^{-(2-1/\beta)})$, we have

$$\frac{\mathrm{d}}{\mathrm{d}P}\mathcal{E}_{B_1 \to B_2}(T_P) > 0.$$

for sufficiently large $D$ and for any $T_P \in [D/B_2, D/B_1]$, thus the optimum satisfies $P^\star = 0$ for sufficiently large $D$. $\qquad\square$

# B EXPERIMENTAL DETAILS AND ADDITIONAL RESULTS

## B.1 LLM PRETRAINING: MODELS, DATA, AND TRAINING SETUP

Unless otherwise specified, language model pretraining in Sections 4 and 5 uses the following settings.

To verify whether the observed phenomena are consistent across scales, we perform experiments under two distinct settings.

Table 1: Model configurations

| Type | LLaMA | | | MoE | |
|---|---|---|---|---|---|
| Model Size | 50M | 200M | 492M | 1001M | 1119M |
| Activated Size | — | — | — | 209M | 291M |
| $d_{\text{model}}$ | 512 | 1024 | 1280 | 512 | 576 |
| $d_{\text{FF}}$ | 2048 | 4096 | 5120 | 1408 | 1152 |
| $d_{\text{FF\_MoE}}$ | — | — | — | 1408 | 192 |
| q_head | 8 | 16 | 20 | 8 | 6 |
| k_head | 8 | 16 | 20 | 4 | 2 |
| depth | 4 | 8 | 15 | 12 | 24 |
| n_expert | — | — | — | 64 | 224 |
| activated_expert | — | — | — | 3 | 16 |

**Small-scale experiment settings.**

- **Model.** LLaMA (Touvron et al., 2023) is a dense, decoder-only Transformer architecture that integrates several modern design components, including Rotary Positional Encoding (RoPE) (Su et al., 2024), Swish-Gated Linear Units (SwiGLU), and Root Mean Square Layer Normalization (RMSNorm). We pretrain LLaMA models with parameter sizes ranging from 50M to 492M. A full list of model configurations is provided in Table 1.

- **Dataset.** Colossal Clean Crawled Corpus (C4) (Raffel et al., 2020) is a large-scale, publicly available language dataset widely adopted for LLM pretraining, including models such as RoBERTa (Liu et al., 2019) and T5 (Raffel et al., 2020). For tokenization, we employ the T5 tokenizer with a vocabulary size of 32,100. Following the setup of Zhao et al. (2024); Zhu et al. (2025); Wang et al. (2025a;b), we train with a sequence length of 256. We use 1,000 linear warmup steps.

**Large-scale experiment settings.**

- **Model.** Shortcut-connected Mixture of Experts (ScMoE) (Cai et al., 2025) is a novel MoE architecture that addresses communication overheads in expert parallelism by introducing shortcut connections and an overlapping parallelization strategy. ScMoE decouples the usual sequential dependency between communication and computation, enabling up to 100% overlap of those two processes, which has demonstrated notable gains in inference efficiency and throughput compared to models of a comparable scale (LongCat et al., 2025). A full list of model configurations is provided in Table 1.

- **Dataset.** We train on a private, real-world LLM dataset to ensure that our experiments closely reflect practical deployment scenarios. The tokenizer is configured with a vocabulary size of 131,072, and training is performed with a maximum sequence length of 8,192.

**Optimizer.** For both small-scale and large-scale experiments, we adopt the standard AdamW (Loshchilov & Hutter, 2019) optimizer as the baseline. The baseline configuration follows protocols from LLaMA pretraining (Touvron et al., 2023), using hyperparameters $\beta_1 = 0.9$, $\beta_2 = 0.95$, weight decay $\lambda = 0.1$, and a gradient clipping threshold of 1.0.

## B.2 LINEAR REGRESSION EXPERIMENTS: SETUP AND DETAILS

We empirically validate that the optimal batch size schedule alone is sufficient to achieve the optimal rates attainable for one-pass SGD in both easy-task and hard-task regimes.

**The easy-task regime.** We consider a task with parameters $s = 1.0$, $\beta = 2.0$, and $\sigma = 1.0$. We set the learning rate $\eta = 0.0005$ and adopt the batch size schedule prescribed by Theorem 3.1 as follows. Recalling that the optimal schedule under the *easy-task* regime satisfies

$$b^{\star}(t) \asymp B_{\max}\big(T^{\star} - t + 1\big)^{\frac{1}{2\beta}-1}, \quad 0 \le t \le T^{\star} \quad \text{with } B_{\max} \asymp D^{\frac{1/2+s\beta}{1+s\beta}}, \; T^{\star} \asymp D^{\frac{\beta}{1+s\beta}}.$$

Due to the discrete nature of batch sizes, we replace the continuous time variable $t$ by the iteration index $k$, and the horizon $T^\star$ by a total number of iterations $K$. We introduce a data-scale hyperparameter $D_0$ and a scale constant $\alpha > 0$ to control the target data scale. The discrete batch size schedule is then constructed as

$$B_k = \left\lfloor D_0^{\frac{1/2+s\beta}{1+s\beta}} (K - k + \nu)^{\frac{1}{2\beta}-1} \right\rfloor, \quad k = 1, \ldots K \quad \text{with } K = \left\lfloor (\alpha D_0)^{\frac{\beta}{1+s\beta}} \right\rfloor.$$

with $\nu > 0$ stabilizes the schedule near the terminal stage. Accordingly, the total data size is of the same order as the data-scale hyperparameter, i.e. $D := \sum_{k=1}^{K} B_k \asymp D_0$. We fix $\alpha = 1000$, $\nu = 10$ and $D_0$ to be 2, 4, 8, 16, and 32. The corresponding values of $D$ are 6346, 13973, 30331, 64962, and 137693. As illustrated in Figure 2 (middle), the batch size schedule alone is sufficient to achieve the minimax optimal risk rate under the *easy-task* regime.

**The hard-task regime.** In this case, we consider the task with $s = 0.4$, $\beta = 2.0$, and $\sigma = 1.0$. We set learning rate $\eta = 0.0005$ and the batch size schedule is configured according to Theorem 3.1 as follows. Recalling that the optimal schedule under the *hard-task* regime satisfies

$$b^\star(t) = \begin{cases} B_{\min}, & 0 \le t < T_1^\star, \\ B_{\max}\big(T^\star - t + 1\big)^{\frac{1}{2\beta}-1}, & T_1^\star \le t \le T^\star, \end{cases}$$

with

$$T^\star \asymp D, \quad \frac{T^\star - T_1^\star}{T^\star} \asymp D^{-\frac{1-1/\beta-s}{2-1/\beta}}, \quad B_{\max} \asymp D^{\frac{s+1}{2}}.$$

Due to the discrete nature of batch sizes, we replace the continuous time variable $t$ by the iteration index $k$, and the horizon $T^\star$ and $T_1^\star$ by a discrete training length $K$ and $K_1$. We introduce introduce a data-scale hyperparameter $D_0$ and a scale constant $\alpha > 0$ to control the target data scale. The discrete batch size schedule is then constructed as

$$B_k = \begin{cases} 1, & \text{for } k = 1, \ldots K_1, \\ \left\lfloor \left(\frac{K-k+\nu}{K-K_1+\nu}\right)^{\frac{1}{2\beta}-1} \right\rfloor & \text{for } k = K_1 + 1, \ldots K, \end{cases}$$

with

$$K = \lfloor \alpha D_0 \rfloor, \quad K_1 = \left\lfloor \alpha(D_0 - D_0^{\frac{s+1}{2-1/\beta}}) \right\rfloor.$$

where scale constant $\nu > 0$ stabilizes the schedule near the terminal stage. Accordingly, the total data size is of the same order as the data-scale hyperparameter, i.e. $D := \sum_{k=1}^{K} B_k \asymp D_0$. We fix $\alpha = 1$, $\nu = 10$ and $D_0$ to be 2000, 4000, 8000, 16000, and 32000. The corresponding values of $D$ are 6346, 13973, 30331, 64962, and 137693. As illustrated by Figure 2 (right), batch size schedule matches the predicted best rate achievable by one-pass SGD with learning rate schedule under the *hard-task* regime.

### B.3 ADDITIONAL DETAILS OF FAST CATCH-UP EXPERIMENTS

We conduct fast catch-up experiments across multiple scales:

- **0.5B model.** We train a 492M ($\approx$ 0.5B) LLaMA model with learning rate $5 \times 10^{-4}$ using a two-stage batch size schedule, switching from 512 to 1024, 2048, 4096 at step 0 and step 25,000 in training, with total 100,000 steps.

- **1B model.** We train a 1001M ($\approx$ 1B) MoE model using a two-stage batch size schedule, switching from 640 to 1280, 2560 at 50B, 200B and 300B tokens in training. In addition, we evaluate a multi-stage schedule that progressively increases the batch size—from 640 to 1280, then 1920, and finally 2560 at 100B, 150B and 200B tokens in training, with total 60,000 steps.

- **1.1B model.** We train a 1119M ($\approx$ 1.1B) MoE model using a two-stage batch size schedule, switching from 1024 to 2048 at 300B and 600B tokens, with total 50,000 steps.

### B.4 ADDITIONAL DETAILS OF SWITCHING-TIME ANALYSIS EXPERIMENTS

In Figure 4 (left), we train a 200M LLaMA model on 10B tokens with learning rate $1 \times 10^{-3}$ using a two-stage batch size schedule, switching from 256 to 512 at different points in training. The total data size corresponding to the full large batch size training step is 30000. We switch batch size at different ratio {0, 1/16, 2/16, 3/16, 4/16, 5/16, 6/16, 7/16, 8/16, 9/16, 10/16, 11/16, 12/16, 13/16, 14/16, 15/16, 16/16}. Each ratio is repeated multiple times to reduce variance in the results.

In Figure 4 (right), we train a 50M LLaMA model on the C4 dataset with learning rate $1 \times 10^{-3}$ , using a small batch size of 128 and a large batch size of 256. The total data sizes corresponding to the full large batch size training step are {20000, 25000, 30000, 35000, 40000, 45000, 50000, 55000, 60000, 65000, 70000, 75000}. For each data size, we perform a grid search to determine the optimal switching point $D^\star$, with a precision of $D/32$. Each configuration of $D^\star/D$ is repeated multiple times to reduce variance in the results.

### B.5 ADDITIONAL DETAILS AND RESULTS FOR LATE-SWITCH SUPERIORITY EXPERIMENTS

We conduct late-switch experiments across multiple scales:

- **0.5B model.** We train a 492M ($\approx$ 0.5B) LLaMA model on 4B tokens with learning rate $5 \times 10^{-4}$ using a two-stage batch size schedule. Specifically, we switch the batch size from 512 to either 1024, 2048, or 4096 at step 25,000.

- **1B model.** We train a 1001M ($\approx$ 1B) MoE model on 0.4T tokens using a two-stage batch size schedule, switching from 640 to either 1280 or 2560 at the 50B, 200B, or 300B token marks. In addition, we evaluate a multi-stage schedule that progressively increases the batch size from 640 to 1280, then 1920, and finally 2560 at 100B, 150B, and 200B tokens, respectively.

- **1.1B model.** We train a 1119M ($\approx$ 1.1B) MoE model on 1T tokens using a two-stage batch size schedule, switching from 1024 to 2048 at either the 300B or 600B token mark.

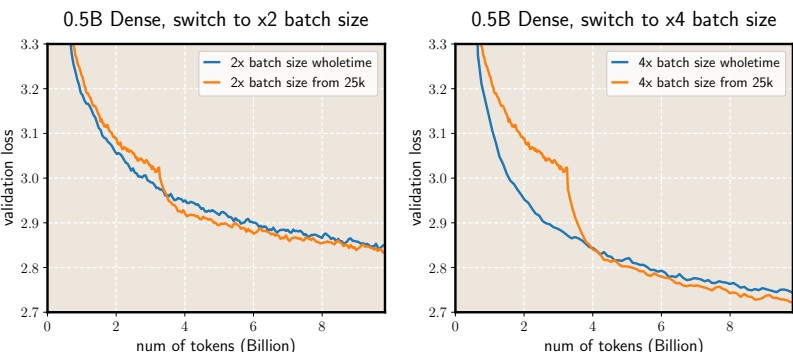

Figure 6: Validation loss versus training tokens under different batch size switching times using 0.5B LLaMA model trained on around 10B tokens, switching batch size from 512 to 1024 (left) and 2048 (right), respectively.

Experimental results are shown in Figure 4 (left), Figure 5, and Figure 6. Moreover, we compare *multi-stage batch size scheduling strategies* for 200M LLaMA model and 1.1B MoE model. For 1119M MoE model, we train on 1T tokens using a four-stage batch size schedule, switching from 1024 to 2048, then 3072 and finally 4096 at different time steps. For 200M LLaMA model, we train on 4B tokens using a three-stage batch size schedule, switching from 128 to 256, then finally 512 at different time steps.

In Figure 7 and Figure 8, the left panels show how batch size evolves with training tokens, while the right panels report the corresponding validation loss. Across both model scales, later switching consistently yields lower validation loss than earlier switching, validating the effectiveness of late-switch superiority in multi-stage batch size scheduling regime.

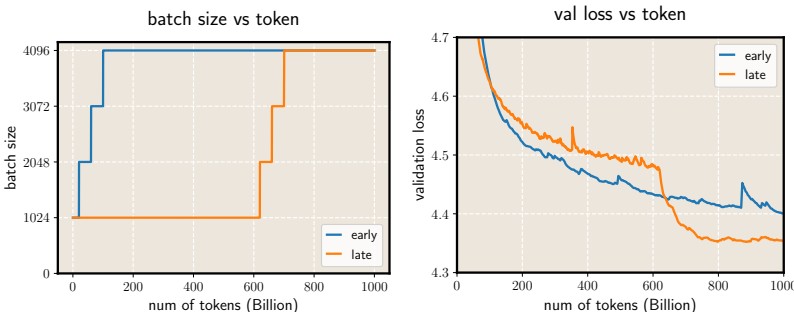

Figure 7: Validation loss versus training tokens with four-stage batch size schedule using 1.1B MoE model trained on 1T tokens. **Left:** batch size versus training tokens; **Right:** validation loss versus training tokens.

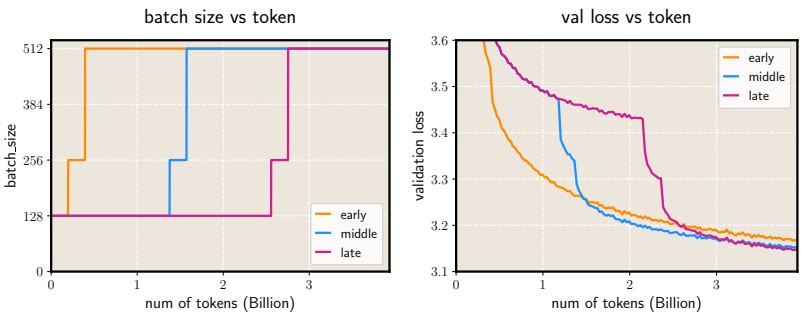

Figure 8: Validation loss versus training tokens with three-stage batch size schedule using 200M LLaMA model trained on 4B tokens. **Left:** batch size versus training tokens; **Right:** validation loss versus training tokens.

## B.6 EXTENSION: INTERACTION WITH LEARNING RATE SCHEDULING

### B.6.1 COMPARISON WITH COSINE AND WSD

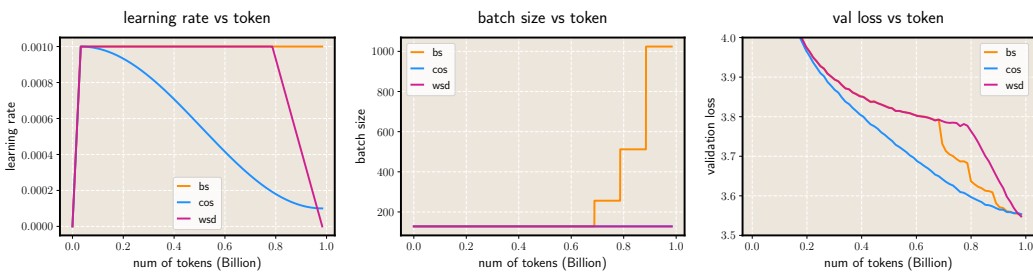

Figure 9: Comparison of validation loss versus training data among batch size schedule, cosine decay learning rate schedule, warmup-stable-decay learning rate schedule using 50M LLaMA model trained on 1B tokens. **Left:** learning rate versus training tokens; **Middle:** batch size versus training tokens; **Right:** validation loss versus training tokens.

We conduct a set of proof-of-concept experiments to evaluate whether a constant learning rate with batch size schedule can perform on par with mainstream learning rate schedulers used in LLM pretraining, such as cosine decay and Warmup–Stable–Decay (WSD) schedule (Hu et al., 2024; Wen et al., 2025). Following established conventions (Hägele et al., 2024), the cosine schedule decays the learning rate to 10% of its maximum value, whereas the WSD schedule decays it to zero, with the ratio of decay phase as 20%. For all figures, the left panels show the evolution of the learning rate over training tokens, the middle panels show the batch size trajectory, and the right

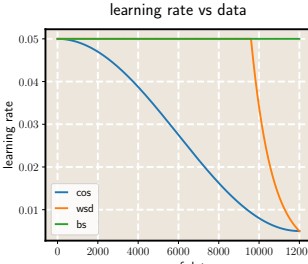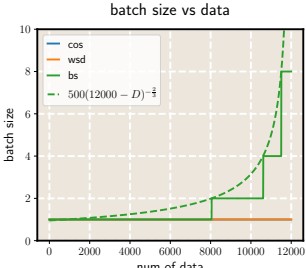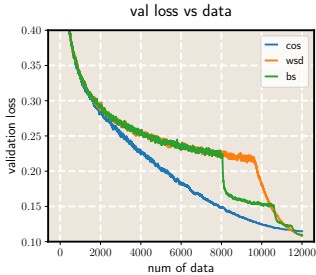

Figure 10: Validation loss versus training tokens among batch size schedule, cosine decay learning rate schedule, warmup-stable-decay learning rate schedule using linear regression model. **Left:** learning rate versus training tokens; **Middle:** batch size versus training tokens; **Right:** validation loss versus training tokens.

panels report the corresponding validation loss curves. We denote the constant learning rate with batch size schedule as 'bs', the cosine schedule as 'cos', and the WSD schedule as 'wsd'.

Figure 9 shows the comparison for LLM pretraining. For the batch size schedule, we begin with a base batch size and increase it in a stage-wise manner: switching to $2\times$ the base batch size at 70% of training tokens, $4\times$ at 80%, and $8\times$ at 90%. The base batch size is set to 128 for the 50M model. We emphasize that this batch size schedule is determined heuristically and is not optimized.

Figure 10 shows the comparison for linear regression. We set $s = 0.3$, $\beta = 1.5$, $\sigma = 2$, $\eta = 0.05$, the exponent in $-2/3$ is the batch size schedule comes from $1/(2\beta) - 1$. With the explicit $\beta$, we design an optimal batch size schedule according to Theorem 3.1.

We observe that, across both LLM pretraining and linear regression, a constant learning rate with an appropriately designed batch size schedule achieves performance comparable to widely adopted learning rate schedulers.

### B.6.2 FAST CATCH-UP UNDER LEARNING RATE DECAY

In this section, to explore the influence of learning rate decay, we replicate the late-switch superiority experiments from Appendix B.5 on 50M and 0.5B LLaMA models using a cosine learning rate schedule. As shown in Figure 11, the characteristic phenomena—fast catch-up and later switching—persist. Note that catch-up is quantified in terms of the intrinsic time $T$. Under a constant learning rate regime, $T$ advances at a uniform pace, whereas a cosine schedule causes it to advance more slowly toward the end of training. Consequently, the apparent merge speed decreases in the final stages. While our current theoretical analysis focuses on a constant LR, the FSL mechanism is general and naturally carries over to other LR schedules, making the theoretical extension to such settings straightforward.

## C STATEMENT

### C.1 ETHICS STATEMENT

We have confirmed that this research was conducted in full compliance with the ICLR Code of Ethics. All experiments respect the principles of integrity, fairness, and transparency. No part of this work involves harm to humans, animals, or the environment, and we have taken care to ensure the responsible use of data, models, and computational resources.

### C.2 REPRODUCIBILITY STATEMENT

We believe that all experimental results in this work are reproducible. The paper specifies comprehensive training and evaluation details—including hyperparameters, optimizer choices, and other relevant settings—in Section 5 and Appendix B. For small-scale experiments, we provide open-source code in the supplemental material, and all datasets used are publicly available. For large-scale

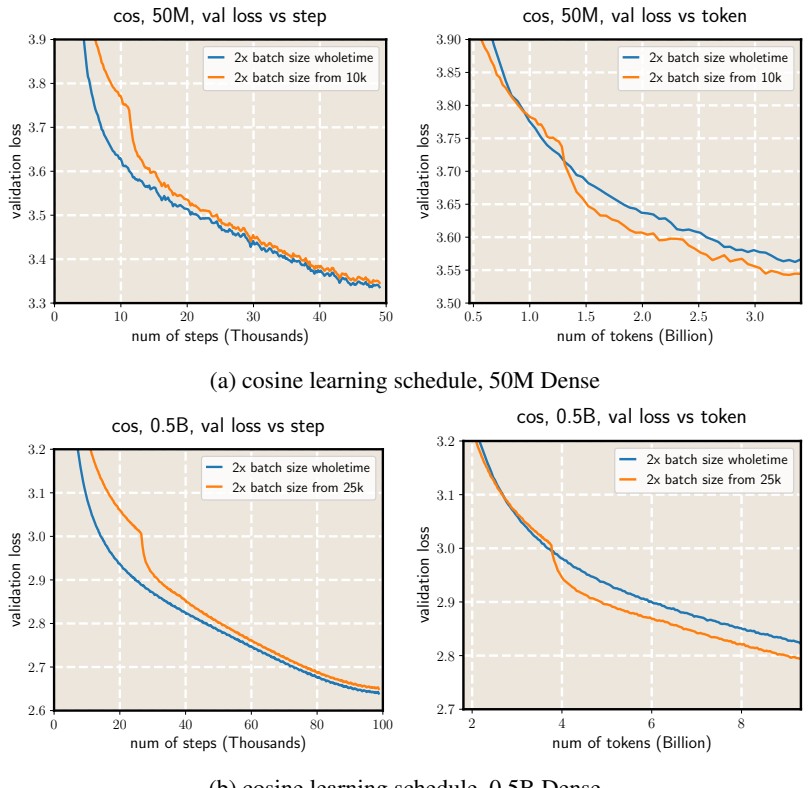

(a) cosine learning schedule, 50M Dense

(b) cosine learning schedule, 0.5B Dense

Figure 11: Two-stage batch size switching using 50M and 0.5B LLaMA model trained on 3.2B and 10B tokens, respectively. **Left:** validation loss versus training steps; **Right:** validation loss versus training tokens.

experiments, we believe that employing comparable datasets and training pipelines will reproduce the same phenomena.

## C.3 LLM USAGE STATEMENT

We used the LLM as a writing assistant during paper preparation. The model was used to identify and correct grammatical errors throughout the manuscript. It suggested ways to make our sentences clearer and smoother. The LLM helped polish the language while keeping our meaning intact. We limited LLM use to only language editing tasks. All research content and ideas came entirely from human work.

Beyond serving as tools, LLMs were themselves the subject of our study. We trained these models and analyzed their behavior to uncover and explain novel phenomena. Importantly, this use of LLMs as research objects should not be misinterpreted as a substantive contribution from the models to the work itself.

