# OpenReview forum: "Fast Catch-Up, Late Switching: Optimal Batch Size Scheduling via Functional Scaling Laws"
_ICLR.cc/2026/Conference — ICLR 2026 Poster_

### Official Review · Reviewer_nk2G · 2025-10-29

**Soundness:** 4
**Presentation:** 4
**Contribution:** 3
**Rating:** 6
**Confidence:** 3

**Summary:**

This paper provides a theoretical foundation for the common practice of batch size scheduling in LLM pre-training. Motivated by two robust empirical observations of a "sudden drop" in loss and a "final merge" of loss curves upon switching batch sizes, the authors apply the Functional Scaling Law (FSL) framework from Li et al. (2025) to explain these dynamics. The theoretical analysis successfully explains these phenomena and yields several key predictions: a "later is better" rule for switching in data-limited regimes, a quantitative power law relating the optimal switch point to the total data budget, and the functional form of a minimax-optimal continuous schedule. These predictions are supported by experiments on models up to 1.1B parameters, bridging the gap between theory and practice.

**Strengths:**

1. **Concrete and Predictive Theory:** This work transforms batch size scheduling from heuristic practice into principled science by applying the FSL framework to derive quantitative predictions. The theorems rigorously characterize loss dynamics (sudden drop and final merge) and provide actionable guidance for schedule design.
2. **Strong Empirical Corroboration:** The theoretical predictions are robustly validated across diverse settings. The core phenomena are shown to be robust across different model architectures (LLaMA and MoE) and scales. It is particularly compelling to see the near-perfect empirical fit to the predicted power-law scaling of optimal switch points, which shows the theory's predictive power and practical utility.

**Weaknesses:**

1. **Constant Learning Rate Assumption:** The analysis assumes a constant learning rate throughout training to isolate batch size effects. This is a major limitation, as modern large-scale pre-training universally employs learning rate schedules (e.g., cosine decay with warmup).
2. **Missing Empirical Validation of the Optimal Schedule:** Theorem 4.4 derives a continuous, increasing batch size schedule that is provably optimal within the FSL framework. This is arguably one of the paper's most significant theoretical contribution, yet it is not empirically tested. The authors mention practical hardware and software constraints that make continuous schedules difficult to implement, but a proof-of-concept using a discretized approximation of the optimal schedule would have strengthened the paper's conclusions.
3. **Asymptotic Nature of the Theory:** The core theoretical results rely on asymptotic analysis where training time or data budget approaches infinity. While experiments demonstrate reasonable approximation quality for large finite runs, the theory cannot formally guarantee performance in the finite regimes where practitioners actually operate.

**Questions:**

1. How should FSL parameters be interpreted for LLMs? The FSL framework relies on a power-law kernel characterized by the capacity parameter $\beta$, which governs eigenvalue decay and drives all theoretical predictions. However, the paper doesn't clarify what $\beta$ represents in language model pre-training. What aspects of the data distribution, architecture, or task does it capture?
2. Can the FSL framework predict optimal joint schedules? Li et al. (2025) used FSL to analyze learning rate schedules, while this paper analyzes batch size schedules. The natural next step is unification: can FSL simultaneously optimize both schedules?

---

> ### Author Response · Authors · 2025-11-25
> **Response to Reviewer nk2G (part 1/3)**
>
> We sincerely thank the reviewer for recognizing the value of our work and for the thoughtful, incisive feedback. We are especially grateful for the reviewer’s generous recognition of our conceptual contributions, the breadth and rigor of our empirical study, and the broader significance of grounding batch-size scheduling within the FSL framework. We respond to the concerns below.
> > **[W1]** Constant Learning Rate Assumption: The analysis assumes a constant learning rate throughout training to isolate batch size effects. This is a major limitation, as modern large-scale pre-training universally employs learning rate schedules (e.g., cosine decay with warmup).
>
> **[A1]** We thank the reviewer for this thoughful question. We would like to clarify following points:
> - **Why we study constant LR?**
>     - **Constant learning rate is important in theory.** Constant learning rate cleanly decouples the effects of batch-size scheduling from those of learning-rate decay, providing a simplified and analytically tractable setting. In a similar vein, works studying learning rate schedule are typically derived under constant batch size as well [1][2][3].
>     - **Constant learning rate is common in practice.** Following the WSD (Warmup-Stable-Decay), a LR schedule which maintains constant LR for the majority of training, notable LLMs such as DeepSeek-V3, MiniMax-01, Kimi-K2 *maintain constant LR for the most of training*. Importantly, their batch-size switching typically occurs during this stable LR phase. Hence, our theoretical setting is close to practice.
> - **Extending to other LR schedule.** To examine whether our conclusions extend beyond the constant-LR case, we conducted additional experiments under a cosine learning-rate schedule in Appendix B.6. The results show that sudden drop, fast merge and later merge **continue to persist**. While our current theoretical analysis focuses on constant LR, the FSL mechanism is general and carries over to other LR schedules; extending the formal theory to such settings is natural.
>
> > **[W2]** Missing Empirical Validation of the Optimal Schedule: Theorem 4.4 derives a continuous, increasing batch size schedule that is provably optimal within the FSL framework. This is arguably one of the paper's most significant theoretical contribution, yet it is not empirically tested. The authors mention practical hardware and software constraints that make continuous schedules difficult to implement, but a proof-of-concept using a discretized approximation of the optimal schedule would have strengthened the paper's conclusions.
>
> **[A2]** We thank the reviewer for this constructive suggestion. In response, we conducted proof-of-concept experiments (Appendix B.5) that a constant LR combined with batch-size schedule performs on par with mainstream LR schedulers in both linear regression and LLM pre-training.
>
> > **[W3]** Asymptotic Nature of the Theory**: The core theoretical results rely on asymptotic analysis where training time or data budget approaches infinity. While experiments demonstrate reasonable approximation quality for large finite runs, the theory cannot formally guarantee performance in the finite regimes where practitioners actually operate.
>
> **[A3]** Most of our central results, such as sudden drop and fast merge, have non-asymptotic forms. The power-law structure of optimal switch points, anyway, does rely on asymptotics. We do want to mention that The TPP (token per parameter, i,e. data vs active model) is typically large in LLM pre-training, e.g., 20 in academic Chincilla law[4], 212 for LLaMA-3 70B[5] and 1098 for Qwen3-32B[6]. In this regime, $D$ and $t$ is large enough that asymptotic analysis should be useful.

---

> ### Author Response · Authors · 2025-11-25
> **Response to Reviewer nk2G (part 2/3)**
>
> > **[Q1]** How should FSL parameters be interpreted for LLMs? The FSL framework relies on a power-law kernel characterized by the capacity parameter
> , which governs eigenvalue decay and drives all theoretical predictions. However, the paper doesn't clarify what represents in language model pre-training. What aspects of the data distribution, architecture, or task does it capture?
>
> **[E1]** We thank the reviewer for this valuable question. Below, we provide a detailed description of the parameters of power-law kernel regression and their interpretation in the context of LLM pre-training.
>
> - **Parameters in Power-Law Kernel Regression**
>
>   - **Model Capacity $\beta$**: A model of the form
>     $$
>     f(\cdot;\boldsymbol{\theta}) = \sum _{j=1}^N \theta_j \phi_j = \sum _{j=1}^N \theta_j \lambda_j^{1/2}  \widehat{\phi}_j \eqsim \sum _{j=1}^N j^{-\beta/2} \widehat{\phi}_j.
>     $$
>
>     shows that higher-index features are increasingly down-weighted by the factor $j^{-\beta/2}$. As $\beta$ increases, the spectrum decays more rapidly, causing the model to **effectively** rely on fewer features. Additionally, for a fixed target function $f^*$, one can use different (potentially non-linear) feature maps $\boldsymbol{\phi}$ (and consequently, different values of $\beta$). The value of $\beta$ reflects the **capacity** of the chosen features. For instance, consider $\boldsymbol{\phi}(\boldsymbol{x}) = \nabla_{\theta}\mathcal{N}(\boldsymbol{x};\theta)$, where $\mathcal{N}(\cdot;\theta)$ denotes a neural network. In this case, $\boldsymbol{\phi}(\boldsymbol{x})$ corresponds to neural tangent features, and the associated kernel
>     $$
>     K_\phi(\boldsymbol{x},\boldsymbol{x}') := \boldsymbol{\phi}(\boldsymbol{x})^\top \boldsymbol{\phi}(\boldsymbol{x}')
>     $$
>     is known as the neural tangent kernel (NTK). Here, the network depth and activation functions govern the spectral decay, determining the effective exponent $\beta$.
>
>   - **Task Difficulty $s$**: The target function admits the expansion
>     $$
>     f^\star =\sum_{j=1}^N  \theta_j^\star \phi_j  \eqsim \sum _{j=1}^N j^{-1/2} \lambda_j^{s/2} \widehat{\phi}_j \eqsim \sum _{j=1}^N  j^{-(s \beta +1)/2}\widehat{\phi}_j.
>     $$
>
>     Since $\{\hat{\phi}_j\}$ are orthonormal, this assumption implies that the spectral energy of $f^\star$ decays according to a power law. The exponent $\alpha:=s\beta$ thus quantifies the task's **intrinsic difficulty**, which depends only on the target function itself and is independent of the model's spectrum. In contrast, $s$ measures the **relative difficulty** with respect to a model of capacity $\beta$: for a fixed $f^*$ (and fixed $\alpha$), adopting a higher-capacity model (smaller $\beta$) increases $s=\alpha/\beta$, making the task relatively easier. In other words, the same task appears easier to a higher-capacity model.
>
> - **Connection Between Power-Law Kernel Regression and LLM Pre-Training** In the context of large language model (LLM) pre-training, the parameter $\beta$ reflects the **model architecture** and determines its capacity. Specifically, $\beta$ is influenced by factors such as the depth of the model, the activation functions, and the choice of feature map. A model with a larger capacity (smaller $\beta$) has a spectrum that decays more slowly, allowing it to utilize a broader range of features, whereas a model with a smaller capacity (larger $\beta$) down-weights higher-index features more rapidly.
> On the other hand, the parameter $s$ reflects the **difficulty of the task** relative to the model architecture. It quantifies how challenging a particular task is for a given model capacity $\beta$. For a fixed target function $f^*$, increasing the model's capacity (reducing $\beta$) leads to a lower value of $s$, making the task easier. In other words, the same task will appear **easier** to a model with a higher capacity, because the model can better accommodate the complexity of the task due to its architecture.
>
>
> > **[Q2]** Can the FSL framework predict optimal joint schedules? Li et al. (2025) used FSL to analyze learning rate schedules, while this paper analyzes batch size schedules. The natural next step is unification: can FSL simultaneously optimize both schedules?
>
> **[E2]** We thank the reviewer for raising this important question. FSL has the potential to simultaneously optimize both schedules. However, both schedules independently achieve the minimax optimal rate; therefore, combining them yields at most **constant-factor** improvements. Establishing these constant-level gains rigorously is *technically non-trivial*, and we view unified LR and batch-size optimization as an exciting direction for future research.

---

> ### Author Response · Authors · 2025-11-25
> **Response to Reviewer nk2G (part 3/3)**
>
> ## **Reference**
>
> [1] [Functional Scaling Laws in Kernel Regression: Loss Dynamics and Learning Rate Schedules](https://openreview.net/forum?id=dpllevHMbc). NeurIPS 2025. \
> [2] [Scaling law with learning rate annealing](https://openreview.net/forum?id=VBx4yMNtjt). NeurIPS 2025. \
> [3] [A multi-power law for loss curve prediction across learning rate schedules](https://openreview.net/forum?id=KnoS9XxIlK). ICLR 2025. \
> [4] [Training Compute-Optimal Large Language Models](https://arxiv.org/abs/2203.15556). NeurIPS 2022. \
> [5] [The llama 3 herd of models](https://arxiv.org/abs/2407.21783). arXiv:2407.21783. \
> [6] [Qwen3 Technical Report](https://arxiv.org/abs/2505.09388). arXiv:2505.09388.

---

> ### Comment · Reviewer_nk2G · 2025-11-28
>
> Thank you for the detailed response. I particularly appreciate new proof-of-concept experiments and the interpretation of parameters of power-law kernel regression in the context of LLM pre-training, which is insightful. I don't have any remaining concerns, while I will keep track of comments from other reviewers. I am happy to increase my score, and I suggest including the discussion on the interpretation of parameters of power-law kernel regression in the revised manuscript.
>
> (__Update:__ Weirdly, I cannot update my rating at this moment for some reason. __I intend to increase my score from 6 to 8__)

---

> ### Author Response · Authors · 2025-11-28
>
> We sincerely thank you for your valuable recommendation, positive feedback, and recognition of the contribution of our work. We will incorporate a discussion on the interpretation of the parameters in power-law kernel regression into the revised manuscript. We are glad that we were able to address your concerns and appreciate your decision to raise the score. We also fully understand the technical issues caused by the recent OpenReview security incident. Thank you again for your time and consideration!

---

### Official Review · Reviewer_A634 · 2025-10-31

**Soundness:** 3
**Presentation:** 3
**Contribution:** 3
**Rating:** 6
**Confidence:** 4

**Summary:**

The paper studies batch size scheduling via a power-law regression theoretical model. It provides an explanation behind two observed effects when switching batch size from low to high: sudden loss drop and eventual matching of the loss curve with the high batch size loss curve. It also proposes an optimal batch size schedule from the theoretical model.

**Strengths:**

1. The paper studies batch size scheduling from a theoretical perspective of a power-law regression model and provides insights related to ‘sudden drop’ and ‘final merge’ of the loss values.

2. It proposes scaling law for optimal switching time from small to large batch in a training run and also empirically verifies that practical settings obey a scaling law.

3. It also proposes an optimal batch size scheduling algorithm for the power-law model.

**Weaknesses:**

1. The paper only studies a constant learning rate schedule, which deviates from practice.

2. Although the paper proposes an optimal batch size scheduling algorithm as a power law, it provides no way of actually developing a practical optimal scheduling algorithm.

3. I don't think Lemma 3 holds for any arbitrary $\theta$, but only for local minimizers as the expected gradient has to be zero for this to hold.

**Questions:**

1. Does Lemma 3 hold generally? Can the authors provide a proof for the same?

2. Is there a practical way of implementing (or obtaining) the optimal batch size scheduling scaling law? Can it be verified that its performance matches the performance of cosine decay in practice?

---

> ### Author Response · Authors · 2025-11-25
> **Response to Reviewer A634**
>
> We sincerely thank the reviewer for recognizing the value of our work. We respond to the concerns below.
> > **[W1]** The paper only studies a constant learning rate schedule, which deviates from practice.
>
> **[A1]** We thank the reviewer for this thoughful question. We would like to clarify following points:
> - **Why we study constant LR?**
>     - **Constant learning rate is important in theory.** Constant learning rate cleanly decouples the effects of batch-size scheduling from those of learning-rate decay, providing a simplified and analytically tractable setting. In a similar vein, works studying learning rate schedule are typically derived under constant batch size as well [1][2][3].
>     - **Constant learning rate is common in practice.** Following the WSD (Warmup-Stable-Decay), a LR schedule which maintains constant LR for the majority of training, notable LLMs such as DeepSeek-V3, MiniMax-01, Kimi-K2 *maintain constant LR for the most of training*. Importantly, their batch-size switching typically occurs during this stable LR phase. Hence, our theoretical setting is close to practice.
> - **Extending to other LR schedule.** To examine whether our conclusions extend beyond the constant-LR case, we conducted additional experiments under a cosine learning-rate schedule in Appendix B.6. The results show that sudden drop, fast merge and later merge **continue to persist**. While our current theoretical analysis focuses on constant LR, the FSL mechanism is general and carries over to other LR schedules; extending the formal theory to such settings is natural.
>
> > **[W2]** Although the paper proposes an optimal batch size scheduling algorithm as a power law, it provides no way of actually developing a practical optimal scheduling algorithm.
> > **[Q2]** Is there a practical way of implementing (or obtaining) the optimal batch size scheduling scaling law? Can it be verified that its performance matches the performance of cosine decay in practice?
>
> **[A2]** We thank the reviewer for this constructive suggestion. In response, we conducted proof-of-concept experiments (Appendix B.5) that a constant LR combined with batch-size schedule performs on par with mainstream LR schedulers in both linear regression and LLM pre-training.
>
> > **[W3]** I don't think Lemma 3 holds for any arbitrary, but only for local minimizers as the expected gradient has to be zero for this to hold.
> > **[Q1]** Does Lemma 3 hold generally? Can the authors provide a proof for the same?
>
> **[A3]** We would like to clarify that this assumption holds generally under our assumption. Our proof for Lemma 3.3 is present in the Appendix A.1.
>
> ## **Reference**
> [1] [Functional Scaling Laws in Kernel Regression: Loss Dynamics and Learning Rate Schedules](https://openreview.net/forum?id=dpllevHMbc). NeurIPS 2025. \
> [2] [Scaling law with learning rate annealing](https://openreview.net/forum?id=VBx4yMNtjt). NeurIPS 2025. \
> [3] [A multi-power law for loss curve prediction across learning rate schedules](https://openreview.net/forum?id=KnoS9XxIlK). ICLR 2025.

---

### Official Review · Reviewer_wCEH · 2025-10-31

**Soundness:** 3
**Presentation:** 3
**Contribution:** 3
**Rating:** 6
**Confidence:** 4

**Summary:**

The paper studies two-stage batch-size schedules for LLM pre-training under a constant learning rate: start with a small batch, then switch once to a larger batch. Empirically, the authors report two robust phenomena across dense and MoE models up to 1.1B parameters and up to 1T tokens: (i) a “sudden drop” in loss at the switch, and (ii) “final merge,” where the loss trajectory measured in steps converges toward the always-large-batch curve. They analyze these behaviors using the Functional Scaling Law (FSL) within a power-law kernel (PLK) teacher–student framework, proving the sudden-drop and final-merge effects, and deriving a “later-switch” rule in data-limited regimes along with a power-law scaling of the optimal switch point with total data size. Experiments at several scales corroborate the theory and show later switches tend to yield better final loss under a fixed token budget.

**Strengths:**

1. Clear empirical phenomena distilled. The paper cleanly isolates and names two behaviors (“sudden drop,” “final merge”) and shows them across architectures/scales, which aids practitioner understanding.
2. Theory that matches practice. The FSL-based analysis explains both phenomena and yields a concrete later-switch rule and a power-law prediction for the optimal switch point, predictions borne out in experiments (including a strong log–log fit).
3. Breadth of evidence. Results span dense (LLaMA-style) and MoE models, billions of tokens, and multiple switch ratios/timings; figures are easy to digest.
4. Actionable takeaway. Under a fixed token budget, “switch later rather than earlier” is a simple guideline practitioners can trial. The paper also discusses hardware constraints motivating staged schedules.

**Weaknesses:**

1. Theory and most experiments assume constant LR, while real LLM training typically uses warmup + cosine/linear decay.
2. Large-scale runs use a private dataset, which limits external reproducibility.

**Questions:**

Do sudden-drop/final-merge and the later-switch rule hold under the cosine decay method?

---

> ### Author Response · Authors · 2025-11-25
> **Response to Reviewer wCEH**
>
> We sincerely appreciate the reviewer's acknowledgement on our clear empirical phenomena and concrete theory. We respond to the concerns below.
>
> > **[W1]** Theory and most experiments assume constant LR, while real LLM training typically uses warmup + cosine/linear decay.
> > **[Q1]** Do sudden-drop/final-merge and the later-switch rule hold under the cosine decay method?
>
> **[A1]** We thank the reviewer for this thoughful question. We would like to clarify following points:
> - **Why we study constant LR?**
>     - **Constant learning rate is important in theory.** Constant learning rate cleanly decouples the effects of batch-size scheduling from those of learning-rate decay, providing a simplified and analytically tractable setting. In a similar vein, works studying learning rate schedule are typically derived under constant batch size as well [1][2][3].
>     - **Constant learning rate is common in practice.** Following the WSD (Warmup-Stable-Decay), a LR schedule which maintains constant LR for the majority of training, notable LLMs such as DeepSeek-V3, MiniMax-01, Kimi-K2 *maintain constant LR for the most of training*. Importantly, their batch-size switching typically occurs during this stable LR phase. Hence, our theoretical setting is close to practice.
> - **Extending to other LR schedule.** To examine whether our conclusions extend beyond the constant-LR case, we conducted additional experiments under a cosine learning-rate schedule in Appendix B.6. The results show that sudden drop, fast merge and later merge **continue to persist**. While our current theoretical analysis focuses on constant LR, the FSL mechanism is general and carries over to other LR schedules; extending the formal theory to such settings is natural.
>
> > **[W2]** Large-scale runs use a private dataset, which limits external reproducibility
>
> **[A2]** Thank you for the comment. We acknowledge that the large-scale runs use a private dataset and cannot be rerun. However, the same phenomenon consistently appears in our public small-scale experiments, suggesting it is not dataset-specific. The use of a real-world private dataset aims to better reflect practical LLM pre-training conditions.
>
> ## **Reference**
> [1] [Functional Scaling Laws in Kernel Regression: Loss Dynamics and Learning Rate Schedules](https://openreview.net/forum?id=dpllevHMbc). NeurIPS 2025. \
> [2] [Scaling law with learning rate annealing](https://openreview.net/forum?id=VBx4yMNtjt). NeurIPS 2025. \
> [3] [A multi-power law for loss curve prediction across learning rate schedules](https://openreview.net/forum?id=KnoS9XxIlK). ICLR 2025.

---

### Official Review · Reviewer_iVq9 · 2025-11-02

**Soundness:** 2
**Presentation:** 2
**Contribution:** 2
**Rating:** 2
**Confidence:** 3

**Summary:**

This paper studies the effect of batch size scheduling in LLM pretraining, focusing on a simplified case, where the batch size increases just once. They make two findings: (1) when the batch size increases, the test error drops quickly then stabilizes, and (2) after a while, the test error curve merges with the one induced by using a constant large batch size from the beginning.

Using a continuous approximation of online SGD applied to kernel linear regression, this paper proves the above two observed phenomena. Moreover, they calculated the optimal batch size schedule (in the simplified sense) using their theory. Finally, they conducted experiments in LLaMA (upto 492M parameters) and MoE models (upto 291M activated parameters) to verify their theory prediction.

**Strengths:**

See below.

**Weaknesses:**

See below.

**Questions:**

My major concern with the paper is regarding the theory component, which seems fabricated instead of fully rigorous. First, the problem setting is highly simplified: constant step size online SGD for kernel linear regression, with a potential two-stage batch size scheduler, and under the source and capacity conditions.

This problem, without the two-stage batch size complication, is very well studied in kernel linear regression literature. However, this paper chooses to take a weird approach by analyzing a continuous approximation of the discrete online SGD process.

Note that in online SGD, the optimization time is tied with the data size. Hence, it can be dangerous extending insights from continuous process to the discrete process. I have noticed one such issue in Theorem 4.4. Detailed as the following question.

1. Theorem 4.4 cannot be correct -- it is well known that one-pass SGD is suboptimal for certain power-law classes. Specifically, in Theorem 4.4, when $\\beta > 1+s\\beta$, the optimal time $T^*$ is greater than data size $D$, which violates the online nature of the algorithm. Therefore, the discussions in Section 4.3 are misleading.

Despite this major issue, it is unclear why the paper focuses on constant step size setting.

In this setting, it is well known that the last iterate of SGD does not converge due to the additive variance error. Indeed, the drop of the loss by increasing batch size is exactly because increasing batch size decreases the variance error. However, in standard SGD literature, one should analyze the averaged iterates or the last SGD iterate but with a decaying step size scheduler. The latter is also what practitioners do. This leads to my second question:

2. For online SGD with a reasonable step size scheduler, e.g., exponentially decaying one or cosine, would increasing batch size also cause a sudden drop of the test error?

Those issues could have been avoided by using the well known rigorous tools developed by prior kernel linear regression literature, instead of using the heuristic continuous approximation.


Besides those theoretical questions, I also feel the experiments are not on a sufficiently large scale. However, I am not an expert here, so I will also wait to see other reviewers’ comments on this.

Overall, I am not fully convinced by the sudden drop of the loss story. But even if I choose to believe it, I do not quite see the implications of this observation:

3. Is the two-stage batch size schedule of any practical importance? When considering more comprehensive batch size schedulers, would the sudden drop still hold to some extent? What's the motivation for studying two-stage batch size schedule?


A minor issue.

4. Lemma 3.3 relies on a certain fourth moment hypercontractivity condition on the feature, however, this is not explicitly mentioned. The constants 2 and 4 seem to suggest they need the feature to be exactly Gaussian.

---

> ### Author Response · Authors · 2025-11-25
> **Response to Reviewer iVq9 (part 1/4)**
>
> We sincerely thank the reviewer for the thoughtful feedback. We are very glad to address the questions and suggestions raised by the reviewer, which we believe will help further refine our work. Below are our responses to the questions and suggestions raised by the reviewer.
>
> >**[Q1]** My major concern with the paper is regarding the theory component, which seems fabricated instead of fully rigorous. First, the problem setting is highly simplified: constant step size online SGD for kernel linear regression, with a potential two-stage batch size scheduler, and under the source and capacity conditions.
>
> **[A1]** We appreciate the reviewer's feedback and will provide clarifications on the following points to address your concerns: the motivation behind our work and the problem setting (one-pass SGD for power-law kernel regression).
>
> - **Motivation of this paper.** First, we would like to emphasize that the primary motivation of our work is **to offer a principled framework for understanding batch size scheduling, rather than to provide rigorous theoretical guarantees**. In practice, LLM pre-training requires not only learning rate scheduling but also batch size scheduling to improve time efficiency [1][2][3]. However, theoretical studies systematically analyzing batch size scheduling are lacking, with existing research primarily offering heuristic approaches, such as increasing the batch size [4][5]. Our goal is to address this gap: *to investigate batch size scheduling from a theoretical standpoint and provide guidance for developing practical batch size scheduling strategies grounded in theoretical insights*. To the best of our knowledge, we are the first study to investigate the impact of batch size scheduling on training dynamics from a theoretical perspective.
> - **Power-law kernel regression setup.** To achieve our goal, we adopt a toy model commonly used in LLM pre-training theory: power-law kernel regression [6][7][8][9][10][11]. The works cited all start with the setup of power-law kernel regression and analyze the associated scaling laws from a theoretical perspective. Specifically, [6] investigates the gradient flow dynamics, [7] examines the behavior of noiseless one-pass SGD with a constant learning rate, [8] explores exponential decay learning rate schedules, [9] focuses on feature learning dynamics, [10] analyzes the multi-pass SGD algorithm in the context of data reuse, and [11] provides a functional scaling law that describes the loss dynamics for general learning rate schedules. However, it is important to note that all of these studies consider the case of constant batch sizes in their analysis. In our work, we follow the functional scaling law framework proposed in [11] and apply it to study batch size scheduling. **To the best of our knowledge, this is the first work to analyze one-pass SGD for kernel regression with batch size scheduling.**
>
> >**[Q2]** This problem, without the two-stage batch size complication, is very well studied in kernel linear regression literature. However, this paper chooses to take a weird approach by analyzing a continuous approximation of the discrete online SGD process.
>
> **[A2]** As the reviewer pointed out, most studies in the kernel regression literature focus on the discrete SGD process, while their works *only provide an upper bound for the last iteration risk or the averaged iteration risk* (such as [12][13][14][15]). However, **understanding the effect of batch size scheduling requires an accurate description of the full loss trajectory**, rather than *just the convergence analysis of the risk at the last iteration or the averaged iteration risk for a class of optimization objectives.* In our work, we consider a specific target $f^*$ and aim to establish pointwise matched upper and lower bounds for the risk dynamics throughout the full training process using SDE modeling and continuous-time analysis. The SDE modeling makes the training dynamics **more tractable**, allowing us to leverage the theoretical framework of functional scaling laws proposed by [11]. Specifically, Theorem 3.4 gives a loss dynamics prediction for arbitrary batch size schedules. As a numerical validation, we empirically demonstrate that the conclusions derived from our SDE model are consistent with experiments using SGD (see Appendix B.7 in the revised version of our paper).

---

> ### Author Response · Authors · 2025-11-25
> **Response to Reviewer iVq9 (part 2/4)**
>
> >**[Q3]** Note that in online SGD, the optimization time is tied with the data size. Hence, it can be dangerous extending insights from continuous process to the discrete process. I have noticed one such issue in Theorem 4.4. Detailed as the following question.
> >> Theorem 4.4 cannot be correct -- it is well known that one-pass SGD is suboptimal for certain power-law classes. Specifically, in Theorem 4.4, when $\beta > 1+s\beta$, the optimal time $T^\star$ is greater than data size $D$, which violates the online nature of the algorithm. Therefore, the discussions in Section 4.3 are misleading.
>
> **[A3]** We sincerely thank the reviewer for this thoughtful suggestion. First, we would like to clarify that our Theorem 4.4 is correct in the context of SDE modeling (see the full proof in the first part of Appendix A.6). To address reviewer's concern, we can further impose a lower-bound condition for the batch size at each time due to the discrete SGD's hardware constraints[16]. This constraint can effectively bridge the gap with discrete SGD, avoiding the infeasible time issue. Formally, for a given data budget $D$ and batch size lower bound $B_{\text{min}}$, the optimal batch size schedule problem can be reformulated as the following functional optimization objective:
>
> $$
> \min_{T, \\\{ B(t)\\\} _{0\leq t\leq T}} \mathcal{E} _{D}  (T, \\\{B(t)\\\} _{0\leq t\leq T}) :=\frac{1}{T^s} + \int _{0}^{T}  \frac{\mathcal{K}(T-t)}{B(t)} \text{d} t.
> $$
>
> $$
> \begin{array}
> \text{s.t.} & \int_{0}^{T} B(t) \text{d} t = D; && \text{(data-constraint)} \\\\
>  & B(t) \ge B_{\text{min}}, \forall 0\leq t \leq T. && \text{(hardware-constraint)}
> \end{array}
> $$
>
> By using KKT condition and analytical variational method, we can derive the following result. The full proof can be found at Appendix A.6.
>
> **Theorem 4.4** (Optimal Batch Size Schedule) For a given data budget $D$ ($D\gg 1$) and the hardware constraint $B_{\operatorname{min}}$, the optimal batch size schedule $B^\star(t)$ ($0\leq t \leq T^\star$) for the above constraint optimization problem takes the following form:
>
> (I) Under the **easy-task** regime ($s > 1 - \frac{1}{\beta}$),
>     $$
>     B^\star(t) \eqsim D^\frac{1+2s\beta}{2+2s\beta}\sqrt{\mathcal{K}(T^*-t)} \eqsim D^\frac{1+2s\beta}{2+2s\beta}(T^\star-t + 1)^{1/2\beta - 1} \text{ for }  0 \le t \le T^\star ,
>     $$ where $T^\star\eqsim D^\frac{\beta}{1+s\beta}$ and optimal excessive risk $\mathcal{E}_{D}^\star \eqsim D^{-\frac{s\beta}{1+s\beta}}.$
>
> (II) Under the **hard-task** regime ($s < 1 - \frac{1}{\beta}$), corresponding to the reviewer's saying $\beta > 1 + s\beta$,
>
> $$
> B^\star(t) = \left\lbrace\begin{array}
> \text{B}_\text{min}, && \text{for } t <  T_1; \\\\
> D^{\frac{s +1}{2}}\sqrt{\mathcal{K}(T^\star-t)} \eqsim D^{\frac{s +1}{2}}(T^\star-t + 1)^{1/2\beta - 1}, && \text{for }  T_1 \le t \le T^\star, \end{array}\right.
> $$
>
> where $T^\star \eqsim D,\; T^\star - T_1 \eqsim D^\frac{s\beta+\beta}{2\beta - 1}$, and optimal excessive risk $\mathcal{E}_{D}^\star \eqsim D^{-s}.$
>
> Theorem 4.4 shows that, under the easy-task regime, the optimal batch size schedule is an increasing schedule with inverse power growth, and the convergence rate of optimal excessive risk $\mathcal{E}_{D} ^\star$ matches the minimax optimal rate under Assumptions 3.1,3.2 (Corollary of Theorem 2 in Caponnetto & De Vito (2007) [17]), suggesting that **a properly designed batch size schedule, even with constant learning rate, is sufficient to attain the minimax optimal rate**. In contrast, under the hard-task regime, the optimal batch size schedule takes the form of stable-growth, which can, to some extent, be considered as *the batch size dual of the warmup-stable-decay (WSD) learning rate schedule* [18][19]. By the hardware constraint $B(t) \ge B_1$, we have $T \lesssim D$ and by FSL $\mathcal{E} _{D}^\star \gtrsim D^{-s} \gtrsim D^{-s\beta/(1+s\beta)}$ in the hard regime. $D^{-s}$ matches the same rate that one-pass SGD with averaged iteration can reach in the hard-task regime [12][13]. Furthermore, We provide a proof-of-concept experiments in Appendix B.5 demonstrating that a constant LR combined with batch-size schedule performs on par with mainstream LR schedulers in both linear regression and LLM pre-training.

---

> ### Author Response · Authors · 2025-11-25
> **Response to Reviewer iVq9 (part 3/4)**
>
> >**[Q4]** Despite this major issue, it is unclear why the paper focuses on constant step size setting.
>
> **[A4]** We thank the reviewer for this thoughful question. We would like to clarify why we focus on constant learning rate as follows:
> - **Constant learning rate is important in theory.** Constant learning rate cleanly decouples the effects of batch-size scheduling from those of learning-rate decay, providing a simplified and analytically tractable setting. In a similar vein, works studying learning rate schedule are typically derived under constant batch size as well [11][20][21].
> - **Constant learning rate is common in practice.** Following the WSD (Warmup-Stable-Decay)[18][19], a LR schedule which maintains constant LR for the majority of training, notable LLMs such as DeepSeek-V3[1], MiniMax-01[22], and Kimi-K2[23] *maintain constant LR for the most of training*. Importantly, their batch-size switching typically occurs during this stable LR phase. Hence, our theoretical setting is close to practice.
> - **Regarding the bias-variance decomposition.** The drop of the loss by increasing batch size cannot simply ascribe to the additive variance error. Different from standard bias and variance decomposition for $\hat{\theta}_t$, our analysis is taken over $\mathbb{E} [\mathcal{R} (\hat{\theta}_t)]$, a bias term connected to noise implicitly via Ito's formula, which differs from your discussion.
>
> >**[Q5]** However, in standard SGD literature, one should analyze the averaged iterates or the last SGD iterate but with a decaying step size scheduler. The latter is also what practitioners do. This leads to my second question:
> >> For online SGD with a reasonable step size scheduler, e.g., exponentially decaying one or cosine, would increasing batch size also cause a sudden drop of the test error?
>
> **[A5]** As the reviewer point out, extending sudden drop to other LR schedule is useful to explore. To examine whether our conclusions extend beyond the constant-LR case, we conducted additional experiments under a cosine learning-rate schedule in Appendix B.6. The results show that sudden drop, fast merge and later merge **continue to persist**. While our current theoretical analysis focuses on constant LR, the FSL mechanism is general and carries over to other LR schedules; extending the formal theory to such settings is natural.
>
> >**[Q6]** Besides those theoretical questions, I also feel the experiments are not on a sufficiently large scale. However, I am not an expert here, so I will also wait to see other reviewers’ comments on this.
>
> **[A6]** We would like to clarify following points. The experiments are **indeed on a sufficiently large scale**. Our experiments are conducted *across different model architectures (LLaMA and MoE) and scales, and reaching 1T tokens with 1.1B model*, a fair scale, which has been acknowledged by other reviewers.
>
> >**[Q7]** Overall, I am not fully convinced by the sudden drop of the loss story. But even if I choose to believe it, I do not quite see the implications of this observation:
> >> Is the two-stage batch size schedule of any practical importance? When considering more comprehensive batch size schedulers, would the sudden drop still hold to some extent? What's the motivation for studying two-stage batch size schedule?
>
> **[A7]** First, the two-stage batch-size schedule serves as a minimal and natural starting point, capturing essential phenomenon without unnecessary complexity—much like how the two-stage lr schedule serves as a starting point for lr schedule study [21]. Second, **Sudden drop and final merge still hold under multiple stage batch size schedule**, as illustated By curve 'multiple' in Figure 1. Moreever, Real-world LLM training typical evolves few stages batch size schedule-e.g.,3-stage (GPT-3, Nematron-4, LLaMA-3) and 5-stage (MiniMax-01), hence *a two-stage schedule is already close to practice*.
>
> >**[Q8]** Lemma 3.3 relies on a certain fourth moment hypercontractivity condition on the feature, however, this is not explicitly mentioned. The constants 2 and 4 seem to suggest they need the feature to be exactly Gaussian.
>
> **[A8]** We would like to clarify that this assumption has been explicitly stated in our setting subsection 3.1 (see the line 168 of our paper). $\epsilon \sim \mathcal{N}(0, \sigma^2)$ and $\boldsymbol{\phi}(x)\sim\mathcal{N}(\boldsymbol{0},\mathbf{H})$. As the reviewer mentioned, Lemma 3.3 can actually extend to a broad class of distribution with fourth moment hypercontractivity condition. Our proof for Lemma 3.3 is present in the Appendix A.1.

---

> ### Author Response · Authors · 2025-11-25
> **Response to Reviewer iVq9 (part 4/4)**
>
> ## **Reference**
>
> [1] [Deepseek-v3 technical report](https://arxiv.org/abs/2412.19437). arXiv:2412.19437. \
> [2] [Nemotron-4 340b technical report](https://arxiv.org/abs/2406.11704). arXiv:2406.11704. \
> [3] [The llama 3 herd of models](https://arxiv.org/abs/2407.21783). arXiv:2407.21783. \
> [4] [An empirical model of large-batch training](https://arxiv.org/abs/1812.06162). arXiv:1812.06162. \
> [5] [Normalization layer per-example gradients are sufficient to predict gradient noise scale in transformers](https://openreview.net/forum?id=S7THlpvH8i). NeurIPS 2024. \
> [6] [A dynamical model of neural scaling laws](https://proceedings.mlr.press/v235/bordelon24a.html). ICML 2024. \
> [7] [4+3 phases of compute-optimal neural scaling laws](https://openreview.net/forum?id=aVSxwicpAk). NeurIPS 2024. \
> [8] [Scaling laws in linear regression: Compute, parameters, and data](https://openreview.net/forum?id=PH7sdEanXP). NeurIPS 2024. \
> [9] [How feature learning can improve neural scaling laws](https://openreview.net/forum?id=dEypApI1MZ). ICLR 2025. \
> [10] [Improved Scaling Laws in Linear Regression via Data Reuse](https://openreview.net/forum?id=jeen4x145W). NeurIPS 2025. \
> [11] [Functional Scaling Laws in Kernel Regression: Loss Dynamics and Learning Rate Schedules](https://openreview.net/forum?id=dpllevHMbc). NeurIPS 2025. \
> [12] [Nonparametric stochastic approximation with large step-sizes](https://doi.org/10.1214/15-AOS1391). Ann. Statist. \
> [13] [Statistical Optimality of Stochastic Gradient Descent on Hard Learning Problems through Multiple Passes](https://proceedings.neurips.cc/paper_files/paper/2018/hash/10ff0b5e85e5b85cc3095d431d8c08b4-Abstract.html). NeurIPS 2018. \
> [14] [Last iterate risk bounds of sgd with decaying stepsize for overparameterized linear regression](https://proceedings.mlr.press/v162/wu22p.html). ICML 2022. \
> [15] [Learning Curves of Stochastic Gradient Descent in Kernel Regression](https://openreview.net/forum?id=U3RtBk95d1). ICML 2025. \
> [16] [Efficient Large-Scale Language Model Training on GPU Clusters Using Megatron-LM](https://doi.org/10.1145/3458817.3476209). SC 2021. \
> [17] [Optimal Rates for the Regularized Least-Squares Algorithm](https://doi.org/10.1007/s10208-006-0196-8). Found Comput Math. \
> [18] [Minicpm: Unveiling the potential of small language models with scalable training strategies](https://openreview.net/forum?id=3X2L2TFr0f). COLM 2024. \
> [19] [Scaling Laws and Compute-Optimal Training Beyond Fixed Training Durations](https://openreview.net/forum?id=Y13gSfTjGr). NeurIPS 2024. \
> [20] [Scaling law with learning rate annealing](https://openreview.net/forum?id=VBx4yMNtjt). NeurIPS 2025. \
> [21] [A multi-power law for loss curve prediction across learning rate schedules](https://openreview.net/forum?id=KnoS9XxIlK). ICLR 2025. \
> [22] [MiniMax-01: Scaling Foundation Models with Lightning Attention](https://arxiv.org/abs/2501.08313). arXiv:2501.08313 \
> [23] [Kimi K2: Open Agentic Intelligence](https://arxiv.org/abs/2507.20534). arXiv:2507.20534.

---

### Comment · Area_Chair_gJnE · 2025-11-28

Dear Reviewers,

The discussion phase is now underway, and the authors have finished uploading their responses to reviewers. If you haven't already, please carefully review the authors' responses to understand their perspectives. Engage in thoughtful, constructive discussions with authors, sharing your thoughts and seeking clarifications. Please also update your review or rating if necessary.

It is noted in the guideline that reviewers can leave comments visible to authors **until Dec 2 11:59pm AoE**. Your active participation and contribution to the ongoing discussion are highly encouraged. Thank you very much for your contribution to ICLR.

Best regards,

AC

---

### Author Response · Authors · 2025-12-03
**Global Response**

Dear AC and reviewers,

We sincerely thank AC and all reviewers for their work during this special incident. Due to this special technical incident, some reviewers have not responded, which we totally understand. Below we provide a brief summary of our work and rebuttal below.

### **Summary**

In LLM pre-training, batch size scheduling is a common parallism strategy for improving training efficiency [1][2][3][4]. However, most existing research on batch size scheduling relies on empirical approaches, with a notable lack of systematic theoretical analysis. To the best of our knowledge, **this is the first work to theoretically investigate batch size scheduling**. Speifcially, we consider a problem setup of one-pass SGD for **power-law kernel regression**, a phenomenological model widely used in the literature of LLM pre-training theory [5][6][7][8][9]. **As the reviewer nk2G said**, “this work transforms batch size scheduling **from heuristic practice into principled science** by applying the FSL framework to derive quantitative predictions.”

### **Contribution**

1. **Phenomena discovery.** Through LLM pre-training experiments, we identify two robust phenomena in batch size switching at the step level: *Sudden drop* in validation loss at the switching point and *Final Merge* of the loss trajectory toward that of the larger batch (cf. Figure 1 in our paper).
2. **Theoretical understanding.** Based on FSL framework [9], we provide a theoretical explanation for these phenomena. Furthermore, our theory reveals a qualitative later-switch rule with a quantitative power-law for optimal batch size switching point, and optimal batch size schedule in our setup.
3. **Extensive experiments.** We empirically validate our theoretical insights through extensive LLM pre-training experiments across diverse architectures (MoE and dense) and scales (up to 1T tokens and 1.1B model parameters), establishing a strong connection between FSL theory and practical batch size scheduling in LLM training.

### **Rebuttal**

- **Reviewer iVq9 (score: 2, conf: 3, no response)** The reviewer's major concern is **our continuous-time SDE modeling and analysis**. However, this approach is widely used in the prior work studying SGD dynamics [9][10][11][12][13], and the reviwer's corresponding qeustion can be resolved by imposing correct hardware constraints. For the remaining concerns about our setup, we have provided detailed clarification and further results.
- **Reviewer wCEH (score: 6, conf: 4, no response)** The reviewer praised our work for **clear empirical phenomena, providing theory that matches practice, broad evidence and offering an actionable takeaway**. To address his concerns, we conducted additional cosine-schedule experiments in Appendix B.6, confirming that our phenomena persist. Our theory can carry over to other LR schedules.
- **Reviewer A634 (score: 6, conf: 4, no response)** The reviewer praised our work for **theoretical insights, our empirically validated power-law for optimal switch time, and optimal batch size schedule**. To address his concerns, we justify our focus on constant LR, provide a proof-of-concept experiment for the optimal batch-size schedule, and clarify the proof of Lemma 3.3.
- **Reviewer nk2G (score: 6→8, conf: 4, response: raising score)** The reviewer praised our work for **concrete and predictive theory "from heuristic practice into principled science" with strong empirical corroboration**. To address his concerns, justify our focus on constant LR, provide a proof-of-concept experiment for the optimal batch-size schedule, explain the rational of asymptotics and FSL parameters. **After discussion, the reviewer intends to increase score from 6 to 8.**

We have updated the paper based on reviewers' suggestion. Thank AC and all reviewers again.

Best regards,

Authors of Submission 18875

---

> ### Author Response · Authors · 2025-12-03
> **Reference for Global Response**
>
> [1] [Deepseek-v3 technical report](https://arxiv.org/abs/2412.19437). arXiv:2412.19437. \
> [2] [Nemotron-4 340b technical report](https://arxiv.org/abs/2406.11704). arXiv:2406.11704. \
> [3] [The llama 3 herd of models](https://arxiv.org/abs/2407.21783). arXiv:2407.21783. \
> [4] [An empirical model of large-batch training](https://arxiv.org/abs/1812.06162). arXiv:1812.06162. \
> [5] [4+3 phases of compute-optimal neural scaling laws](https://openreview.net/forum?id=aVSxwicpAk). NeurIPS 2024. \
> [6] [Scaling laws in linear regression: Compute, parameters, and data](https://openreview.net/forum?id=PH7sdEanXP). NeurIPS 2024. \
> [7] [How feature learning can improve neural scaling laws](https://openreview.net/forum?id=dEypApI1MZ). ICLR 2025. \
> [8] [Improved Scaling Laws in Linear Regression via Data Reuse](https://openreview.net/forum?id=jeen4x145W). NeurIPS 2025. \
> [9] [Functional Scaling Laws in Kernel Regression: Loss Dynamics and Learning Rate Schedules](https://openreview.net/forum?id=dpllevHMbc). NeurIPS 2025. \
> [10] [Stochastic modified equations and dynamics of stochastic gradient algorithms i: Mathematical foundations](https://jmlr.csail.mit.edu/papers/volume20/17-526/17-526.pdf). JMLR. \
> [11] [What Happens after SGD Reaches Zero Loss?--A Mathematical Framework](https://openreview.net/forum?id=siCt4xZn5Ve). ICLR 2022. \
> [12] [On the SDEs and Scaling Rules for Adaptive Gradient Algorithms](https://openreview.net/forum?id=F2mhzjHkQP). NeurIPS 2022. \
> [13] [Learning Rate Schedules in the Presence of Distribution Shift](https://proceedings.mlr.press/v202/fahrbach23a/fahrbach23a.pdf). ICML 2023.

---

### Meta-Review · Area_Chair_2qvf · 2026-01-07

**Summary:**

Three reviewers recommend acceptance, but reviewer IVQ9 is sceptical about the theory aspect.

I recommend that the authors carefully consider the arguments of IVQ9 in the camera-ready version.

**Reviewer Concerns:**

Most reviewers had their concerns clarified, except IVQ9, who believes the theory aspect is weakly positioned. However, the authors provided a thorough response to the reviewer.

**Reviewer Scores:**

nk2G would increase the score to 8

---

### Decision · Program_Chairs · 2026-01-26

Accept (Poster)